# Rotation-Invariant Random Features Provide a Strong Baseline for Machine Learning on 3D Point Clouds

**Owen Melia**                                                                    *meliao@uchicago.edu*
*Department of Computer Science*
*University of Chicago*

**Eric Jonas**                                                                       *ericj@uchicago.edu*
*Department of Computer Science*
*University of Chicago*

**Rebecca Willett**                                                              *willett@uchicago.edu*
*Department of Computer Science and Statistics*
*University of Chicago*

**Reviewed on OpenReview:** *https://openreview.net/forum?id=nYzhlFyjjd*

## Abstract

Rotational invariance is a popular inductive bias used by many fields in machine learning, such as computer vision and machine learning for quantum chemistry. Rotation-invariant machine learning methods set the state of the art for many tasks, including molecular property prediction and 3D shape classification. These methods generally either rely on task-specific rotation-invariant features, or they use general-purpose deep neural networks which are complicated to design and train. However, it is unclear whether the success of these methods is primarily due to the rotation invariance or the deep neural networks. To address this question, we suggest a simple and general-purpose method for learning rotation-invariant functions of three-dimensional point cloud data using a random features approach. Specifically, we extend the random features method of Rahimi & Recht (2007) by deriving a version that is invariant to three-dimensional rotations and showing that it is fast to evaluate on point cloud data. We show through experiments that our method matches or outperforms the performance of general-purpose rotation-invariant neural networks on standard molecular property prediction benchmark datasets QM7 and QM9. We also show that our method is general-purpose and provides a rotation-invariant baseline on the ModelNet40 shape classification task. Finally, we show that our method has an order of magnitude smaller prediction latency than competing kernel methods.

## 1 Introduction

Many common prediction tasks where the inputs are three-dimensional physical objects are known to be rotation-invariant; the ground-truth label does not change when the object is rotated. Building rotation invariance into machine learning models is an important inductive bias for such problems. The common intuition is that restricting the learning process to rotation-invariant models will remove any possibility of poor generalization performance due to rotations of test samples and may improve sample efficiency by reducing the effective complexity of learned models. These ideas have inspired a line of research begun by Kondor et al. (2018); Cohen et al. (2018a); Esteves et al. (2018) into building general-purpose deep neural network architectures that are invariant to rotations of their input. However, in these studies, it is not clear whether the reported high accuracies are due to the expressive power of the neural networks or primarily attributable to rotation invariance. We introduce a rotation-invariant random feature model which helps us explore the impact of rotation invariance alone, outside of the neural network framework. Our method is general-purpose

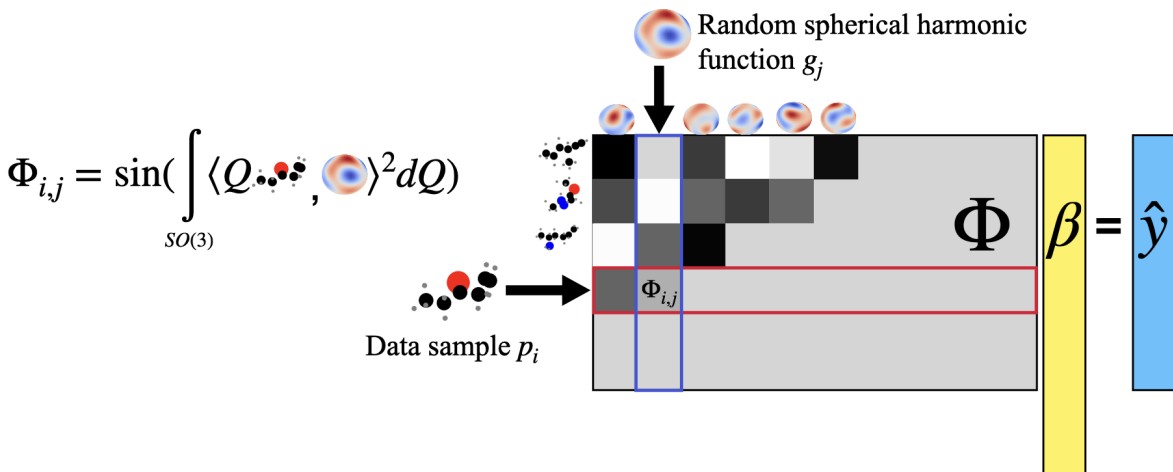

Figure 1: An overview of the rotation-invariant random features method. First we construct a feature matrix $\Phi$, defined in Equation (7). The rows of $\Phi$ correspond to $p_i$, different samples in our training set, and the columns of $\Phi$ correspond to $g_j$, different random sums of spherical harmonic functions, which are defined in Equation (10). The entries of the feature matrix are rotation-invariant random features. To compute the random features, we analytically evaluate the integral over all possible rotations of the data sample $p_i$. We describe our method for evaluating the random features in Section 4.1. After constructing the feature matrix, we fit a set of linear weights $\beta$ and make predictions by evaluating the linear model $\Phi\beta = \hat{y}$.

and does not require expert knowledge for feature or architecture design. For certain prediction tasks, our proposed method can be computed with very low prediction latency with only a small reduction in accuracy compared to neural network methods, making it a viable method in a range of applications.

In this paper, we consider prediction problems that are *rotation-invariant*, that is, the ground-truth response $f^*(x) = y$ does not change when an arbitrary rotation is applied to the input $x$. We represent the input data $x$ as a 3D point cloud, an unordered set of points in $\mathbb{R}^3$ possibly with accompanying labels. Common examples of rotation-invariant prediction problems on 3D point clouds include molecular property prediction, where the point cloud consists of the positions of a molecule's constituent atoms, and 3D shape classification, where the points are sampled from the surface of an object. Section 2 provides a detailed overview of methods used for rotation-invariant prediction, their underlying insights, and their various advantages and disadvantages.

We propose a new method for learning rotation-invariant functions of point cloud data, which we call rotation-invariant random features. We extend the random features method of Rahimi & Recht (2007) by deriving a version that is invariant to three-dimensional rotations and showing that it is fast to evaluate on point cloud data. The rotation invariance property of our features is clear from their definition, and computing the features only requires two simple results from the representation theory of $SO(3)$. Despite its simplicity, our method achieves performance near state-of-the-art on small-molecule energy regression and 3D shape classification. The strong performance on both tasks gives evidence that our method is general-purpose. An overview of our method can be found in Figure 1.

Our experiments in Section 5.1.1 show that our model is an order of magnitude faster than state-of-the-art kernel methods when predicting on new samples for energy regression tasks. Thus we conclude our model is a promising candidate for tasks which require real-time rotation-invariant predictions. Minimizing prediction latency is a growing concern for many machine learning methods; there are multiple applications requiring low-latency predictions on point clouds. Using machine learning predictions for real-time system control places hard requirements on prediction latencies. For example, trigger algorithms in the ATLAS experiment at the CERN Large Hadron Collider require prediction latencies ranging from $2\mu s$ to $40ms$ at different stages of the event filtering hierarchy (Collaboration, 2008). The input to these trigger algorithms is a particle jet, which can be interpreted as a point cloud in four-dimensional spacetime. The majority of data-driven trigger

algorithms either impose rotational invariance in phase space (Komiske et al., 2018; Thaler & Van Tilburg, 2011) or enforce invariance to the Lorentz group, which includes three-dimensional rotations and relativistic boosts (Bogatskiy et al., 2022; Gong et al., 2022). These definitions of invariance are compatible with our framework, with slight adjustments. Another use-case for low-latency models is as replacements for force fields in molecular dynamics simulations (Gilmer et al., 2017). These calculations make serial subroutine calls to force field models, meaning latency improvements of these force field models are necessary to speed up the outer simulations. Other rotation-invariant machine learning models either rely on deep computational graphs, which provide a fundamental limit to speedup via parallelization, or they rely on kernel methods (Christensen et al., 2020) which have prediction-time complexity linear in the number of training samples.

## 1.1 Contributions

- We derive a rotation-invariant version of the standard random features approach presented in Rahimi & Recht (2007). By using simple ideas from the representation theory of $SO(3)$, our rotation-invariant random feature method is easy to describe and implement, and requires minimal expert knowledge in its design (Section 4).

- We show with experiments that our rotation-invariant random feature method outperforms or matches the performance of general-purpose equivariant architectures optimized for small-molecule energy regression tasks. In particular, we outperform or match the performance of Spherical CNNs (Cohen et al., 2018a) on the QM7 dataset and Cormorant (Anderson et al., 2019) on the QM9 dataset. We compare our test errors with state-of-the-art methods on the QM9 dataset to quantify the benefit of designing task-specific rotation invariant architectures (Section 5.1).

- We show that our method makes predictions an order of magnitude faster than kernel methods at test time (Section 5.1.1).

- We show the rotation-invariant random feature method is general-purpose by achieving low test errors using the same method on a different task, shape classification on the ModelNet40 benchmark dataset (Section 5.2).

- We make our code publicly available at
  `https://github.com/meliao/rotation-invariant-random-features`.

## 2 Related Work

In this section, we describe two major types of rotation-invariant prediction methods. The first class of methods extracts rotation-invariant features using expert chemistry knowledge. The second class of methods uses deep neural networks to build general-purpose invariant architectures. Finally, we briefly survey the random features work that inspire our method.

**Descriptors of Atomic Environments**   In the computational chemistry community, significant effort has been invested in using physical knowledge to generate "descriptors of atomic environments," or rotation-invariant feature embeddings of molecular configurations. One such feature embedding is the Coulomb matrix, used by Montavon et al. (2012). In Rupp et al. (2012), the sorted eigenvalues of the Coulomb matrix are used as input features. The sorted eigenvalues are invariant to rotations of the molecule and give a more compact representation than the full Coulomb matrix. Noticing that the Coulomb matrix can relatively faithfully represent a molecule with only pairwise atomic interactions has been an important idea in this community. Many methods have been developed to generate descriptions of atomic environments by decomposing the representation into contributions from all atomic pairs, and other higher-order approximations consider triplets and quadruplets of atoms and beyond (Christensen et al., 2020; Bartók et al., 2013; Kovács et al., 2021; Drautz, 2019).[1]  The FCHL19 (Christensen et al., 2020) method creates a featurization that relies on

---

[1]Many general-purpose invariant methods follow this design as well. Cormorant (Anderson et al., 2019) has internal nodes which correspond to all pairs of atoms in the molecule, and Spherical CNNs (Cohen et al., 2018a) render spherical images from molecular point clouds using ideas from pairwise electrostatic (Coulombic) repulsion.

chemical knowledge and is optimized for learning the energy of small organic molecules. The FCHL19 feature embedding depends on 2-body and 3-body terms. There are multiple different approaches, such as Smooth Overlap of Atomic Potentials (SOAP) (Bartók et al., 2013) and Atomic Cluster Expansion (ACE) (Drautz, 2019; Kovács et al., 2021) which first compute a rotation-invariant basis expansion of a chemical system, and then learn models in this basis expansion. In Appendix B, we show that our proposed method can be interpreted as using a highly-simplified set of ACE basis functions to learn a randomized nonlinear model.

Deep learning is an effective tool in computational chemistry as well. Task-specific deep neural network architectures for learning molecular properties of small organic molecules outperform other approaches on common energy learning benchmarks. Multiple methods (Schütt et al., 2018; Unke & Meuwly, 2019) design neural networks to take atomic charges and interatomic distances as input. These methods use highly-optimized neural network architectures, including specially-designed atomic interaction blocks and self-attention layers. OrbNet (Qiao et al., 2020), another neural network method, takes the output of a low-cost density functional theory calculation as rotation-invariant input features and uses a message-passing graph neural network architecture.

**General-Purpose Invariant Architectures**  While many general-purpose invariant architectures operate directly on point clouds, the first examples of such architectures, Spherical CNNs (Cohen et al., 2018a; Esteves et al., 2018; Kondor et al., 2018) operate on "spherical images", or functions defined on the unit sphere. These networks define spherical convolutions between a spherical image and a filter. These convolutions are performed in Fourier space, and the input spherical image is transformed using a fast Fourier transform on the unit sphere. Much like a traditional convolutional neural network, these networks are comprised of multiple convolutional layers in series with nonlinearities and pooling layers in between. To accommodate multiple different types of 3D data, including 3D point clouds, these methods take a preprocessing step to render the information of a 3D object into a spherical image.

Other methods operate on point clouds directly. SPHNet (Poulenard et al., 2019) computes the spherical harmonic power spectrum of the input point cloud, which is a set of rotation-invariant features, and passes this through a series of point convolutional layers. Cormorant (Anderson et al., 2019) and Tensor Field Networks (Thomas et al., 2018) both design neural networks that at the first layer have activations corresponding to each point in the point cloud. All activations of these networks are so-called "spherical tensors," which means they are equivariant to rotations of the input point cloud. To combine the spherical tensors at subsequent layers, Clebsch-Gordan products are used. Many other deep neural network architectures have been designed to operate on point clouds, but the majority have not been designed to respect rotational invariance. A survey of this literature can be found in Guo et al. (2021).

There are many ways to categorize the broad field of group-invariant deep neural network architectures, and we have chosen to form categories based on the input data type. Another helpful categorization is the distinction between *regular* group-CNNs and *steerable* group-CNNs introduced by Cohen et al. (2018b). In this distinction, *regular* group-CNNs form intermediate representations that are scalar functions of the sphere or the rotation group, and examples of this category are Cohen et al. (2018a); Kondor et al. (2018); Poulenard et al. (2019). *Steerable* group-CNNs form intermediate representations that are comprised of "spherical tensors", and examples of this category are Anderson et al. (2019); Weiler et al. (2018); Thomas et al. (2018).

More recently, some works have investigated methods that endow non-invariant point cloud neural network architectures with a group-invariance property, either through a modification of the architecture or a data preprocessing step. Xiao et al. (2020) suggests using a preprocessing step that aligns all point clouds in a canonical rotation-invariant alignment; Puny et al. (2022) expands this idea to general transformation groups. After the preprocessing step, any point cloud neural network architecture can be used to make rotation-invariant predictions of the input. Vector Neurons (Deng et al., 2021) endows neural network architectures with rotation equivariance by using three-dimensional vector activations for each neuron in the network and cleverly modifying the equivariant nonlinearities and pooling. These methods have been successfully applied to dense point clouds that arise in computer vision problems such as shape classification and segmentation. Molecular point clouds are qualitatively different; they are quite sparse, and the Euclidean nearest neighbor graph does not always match the graph defined by molecular bond structure. Because of these differences, the alignment methods and network architectures designed for solving shape classification

and segmentation tasks are not *a priori* good models for learning functions of molecular point clouds. To the best of our knowledge, they have not been applied successfully in molecular property prediction tasks. Finally, Villar et al. (2021) show that any rotation-invariant function of a point cloud can be expressed as a function of the inner products between individual points; this insight reduces the problem of designing rotation-invariant models to one of designing permutation-invariant architectures.

**Random Features**   Random Fourier features (Rahimi & Recht, 2007) are an efficient, randomized method of approximating common kernels. For kernel methods, prediction time scales linearly with the size of the dataset. Random Fourier feature methods require time linear in $d$, an approximation parameter. The approximation works by drawing a random vector $\xi$ and defining a low-dimensional feature embedding $\varphi(x; \xi)$, called a random Fourier feature. The random Fourier features approximate a shift-invariant kernel $k(x, x') = k(x - x')$ in the sense that the inner product of two random Fourier feature evaluations is an unbiased estimate of the kernel:

$$\mathbb{E}_\xi \left[ \langle \varphi(x; \xi), \varphi(x'; \xi) \rangle \right] = k(x, x')$$

This approximation holds when the random vectors $\xi$ are drawn from the Fourier transform of the kernel $k$ and the random Fourier features have the following functional form:

$$\varphi(x; \xi) = [\cos(\langle x, \xi \rangle), \quad \sin(\langle x, \xi \rangle)]^\top$$

To reduce the variance of this approximation, one can draw $d$ such random vectors $\{\xi_1, ..., \xi_d\}$ and concatenate the resulting random features. As a practical method, Rahimi & Recht (2007) proposes building a feature matrix $\Phi \in \mathbb{R}^{n \times d}$ by drawing $d$ different random Fourier features $\varphi(\,\cdot\,; \xi_j)$ and evaluating the random Fourier features at each of the $n$ samples $x_i$ in the training dataset:

$$\Phi_{i,j} = \varphi(x_i; \xi_j)$$

Once the feature matrix is formed, a linear model is trained to fit a response vector $y$ using regularized linear regression, such as ridge regression:

$$\underset{\beta}{\arg\min} \, \|\Phi\beta - y\|_2^2 + \lambda\|\beta\|_2^2 \tag{1}$$

Experiments in Rahimi & Recht (2007) show this is an effective method for fitting data, even when $d \ll n$. When $d \ll n$, the model can be evaluated much faster than the original kernel.

Follow-up work, including Rahimi & Recht (2008), suggested that interpreting random feature methods as kernel approximators is not necessary. This work uses random nonlinear features with varying functional forms, including random decision stumps and randomly-initialized sigmoid neurons $\sigma(\langle \xi, x \rangle)$. These random nonlinear features form effective models for diverse types of data. In Section 4, we introduce random features similar to those in Rahimi & Recht (2007). Our method does not approximate any explicit kernel, and it is designed to be invariant to any rotation of the input data, so we call our method rotation-invariant random features.

Finally, the notion of group-invariant random feature models also appears in Mei et al. (2021) as a technical tool to understand the sample complexity benefit of enforcing group invariances in overparameterized models. In this work, the invariances considered are transformations on one-dimensional signals that arise from cyclic and translation groups.

## 3   Rotational Invariance and Spherical Harmonics

In this section, we will introduce rotational invariance and the spherical harmonics. In our method, we use the definitions presented in this section and only two simple facts about the spherical harmonics and rotations, which we list at the end of this section. We say a function $f$ mapping a point cloud $p$ to an element of the vector space $\mathcal{Y}$ is rotation-equivariant if there exists some group action on the vector space $\mathcal{Y}$ such that

$$f(Q \circ p) = Q \circ f(p) \quad \forall Q \in SO(3)$$

Intuitively, this means that when an input to $f$ is rotated, $f$ preserves the group structure and the output changes predictably. We say that a function $f$ is rotation-invariant if the group action on the output vector space $\mathcal{Y}$ is the identity:

$$f(p) = f(Q \circ p) = f(\{Qx_1, Qx_2, \ldots, Qx_N\}) \quad \forall Q \in SO(3) \tag{2}$$

Finally, our goal is to learn a function over point clouds that does not change when any rotation is applied to the point cloud. Formally, given some distribution $\mathcal{D}$ generating pairs of data $(p, y)$ our learning goal is to find

$$f^* = \arg\min_f \mathbb{E}_{(p,y) \sim \mathcal{D}} \left[ l(f(p), y) \right]$$

where $l$ is a classification or regression loss depending on the task and the minimization is over all functions that satisfy a rotational invariance constraint Equation (2).

### 3.1 Spherical Harmonics

When decomposing a periodic function $f : [0, 2\pi] \to \mathbb{R}$, a natural choice of basis functions for the decomposition is the complex exponentials $e^{imx}$. In this basis, $f(x)$ may be expressed as $f(x) = \sum_m a_m e^{imx}$ where $a_m = \langle f(x), e^{imx} \rangle$. When decomposing a function on the unit sphere $S^2$, a similar set of basis functions emerges, and they are called the spherical harmonics. The complex exponentials are indexed by a single parameter $m$, and because the spherical harmonics span a more complicated space of functions, they require two indices, $\ell$ and $m$. By convention, the indices are restricted to $\ell \geq 0$ and $-\ell \leq m \leq \ell$. A single spherical harmonic function $Y_m^{(\ell)}$ maps from the unit sphere $S^2$ to the complex plane $\mathbb{C}$. For a function $f(x) : S^2 \to \mathbb{R}$ we can compute an expansion in the spherical harmonic basis:

$$f(x) = \sum_{m,\ell} a_m^{(\ell)} Y_m^{(\ell)}(x)$$
$$a_m^{(\ell)} = \langle f(x), Y_m^{(\ell)}(x) \rangle$$

In designing our method, we use two simple facts about spherical harmonics:

- If one has evaluated the spherical harmonics of some original point $x \in S^2$ and wants to evaluate those spherical harmonics at a new rotated point $Qx$ for some rotation matrix $Q$, the rotated evaluation is a linear combination of the un-rotated spherical harmonics at the same index $\ell$:

$$Y_m^{(\ell)}(Qx) = \sum_{m'=-\ell}^{\ell} Y_{m'}^{(\ell)}(x) D^{(\ell)}(Q)_{m,m'} \tag{3}$$

  Here $D^\ell(Q)$ is a Wigner-D matrix; it is a square matrix of size $(2\ell + 1 \times 2\ell + 1)$.

- Evaluating the inner product between two elements of the Wigner D-matrices is simple. In particular, when $dQ$ is the uniform measure over $SO(3)$, we have the following expression:

$$\int_{SO(3)} D^{(\ell_1)}(Q)_{m_1,k_1} D^{(\ell_2)}(Q)_{m_2,k_2} dQ = (-1)^{m_1-k_1} \frac{8\pi^2}{2\ell+1} \delta_{-m_1,m_2} \delta_{-k_1,k_2} \delta_{\ell_1,\ell_2} \tag{4}$$

  This is a corollary of Schur's lemma from group representation theory, and we discuss this fact in Appendix C.

## 4 Rotation-Invariant Random Features

In our setting, an individual data sample is a point cloud $p_i = \{x_{i,1}, \ldots, x_{i,N_i}\}$ with points $x_{i,j} \in \mathbb{R}^3$. We model the data as a function: $p_i(x) = \sum_{j=1}^{N_i} \delta(x - x_{i,j})$ where $\delta(\cdot)$ is a delta function centered at 0. We can

then use a functional version of the random feature method where our data $p_i$ is a function, $g$ is a random function drawn from some pre-specified distribution, and $\langle \cdot, \cdot \rangle$ is the $L^2$ inner product:

$$\tilde{\varphi}(p_i; g) = \sin\left(\langle p_i, g \rangle\right) \tag{5}$$

We want the random feature to remain unchanged after rotating the point cloud, but Equation (5) does not satisfy this. For general functions $g$, $\langle Q \circ p_i, g \rangle \neq \langle p_i, g \rangle$. We achieve rotational invariance by defining the following rotation-invariant random feature:

$$\varphi(p_i; g) = \sin\left(\int_{SO(3)} \langle Q \circ p_i, g \rangle^2 dQ\right) \tag{6}$$

We "symmetrize" the inner product by integrating over all possible orientations of the point cloud $p_i$, eliminating any dependence of $\varphi(\cdot; g_j)$ on the data's initial orientation. Because $\varphi(\cdot; g_j)$ does not depend on the initial orientation of the data, it will not change if this initial orientation changes, i.e., if $p_i$ is rotated by $Q$. To build a regression model, we construct a feature matrix

$$\Phi_{i,j} = \varphi(p_i; g_j) \tag{7}$$

and fit a linear model to this feature matrix using ridge regression, as in Equation (1).

## 4.1 Evaluating the Random Features

In this section, we describe how to efficiently evaluate the integral in Equation (6). $SO(3)$ is a three-dimensional manifold and the integral has a nonlinear dependence on the data, so methods of evaluating this integral are not immediately clear. However, representation theory of $SO(3)$ renders this integral analytically tractable and efficiently computable. The main ideas used to evaluate this integral are exploiting the linearity of inner products and integration, and choosing a particular distribution of random functions $g$. First, we observe that because the rotated data function $Q \circ p_i$ is the sum of a few delta functions, we can expand the inner product as a sum of evaluations of the random function $g$:

$$\langle Q \circ p_i, g \rangle = \sum_{j=1}^{N_i} g(Q x_{i,j}) \tag{8}$$

Expanding the inner product and the quadratic that appear in Equation (6) and using the linearity of the integral, we are left with a sum of simpler integrals:

$$\int_{SO(3)} \langle Q \circ p_i, g \rangle^2 dQ = \sum_{j_1, j_2=1}^{N_i} \int_{SO(3)} g(Q x_{i,j_1}) g(Q x_{i,j_2}) dQ \tag{9}$$

Next, we choose a particular distribution for $g$ which allows for easy integration over $SO(3)$. We choose to decompose $g$ as a sum of randomly-weighted spherical harmonics with maximum order $L$ and $K$ fixed radial functions:

$$g(x) = \sum_{k=1}^{K} \sum_{\ell=0}^{L} \sum_{m=-\ell}^{\ell} w_{m,k}^{(\ell)} Y_m^{(\ell)}(\hat{x}) R_k(\|x\|) \tag{10}$$

where $w_{m,k}^{(\ell)}$ are random weights drawn from a Gaussian distribution $\mathcal{N}(0, \sigma^2)$, $\hat{x} = \frac{x}{\|x\|}$, and $R_k : \mathbb{R}_+ \to \mathbb{R}$ is a radial function. Two examples of such radial functions are shown in Figure 2. This definition of random functions gives us a few hyperparameters; $L$, the maximum frequency of the spherical harmonics; $\sigma$, the standard deviation of the randomly-drawn weights; and the choice of radial functions. In Appendix E, we show that the performance of our method is most sensitive to $\sigma$.

By repeated application of the linearity of the integral and the two facts about the spherical harmonics mentioned in Section 3.1, we are able to write down a closed-form expression for the integral on the left-hand side of Equation (9). We include these algebraic details in Appendix A. Computing this integral has relatively low complexity; it requires evaluating a table of spherical harmonics for each point $x_{i,j}$ up to maximum order $L$, and then performing a particular tensor contraction between the array of spherical harmonic evaluations and the array of random weights. This tensor contraction has complexity $O(N^2 L^3 K^2)$.

## 5 Experiments

We conduct multiple experiments comparing our method with other landmark rotation-invariant machine learning methods. We find that for predicting the atomization energy of small molecules, our method outperforms neural networks in the small dataset setting, and we have competitive test errors in the large dataset setting. Our method is an order of magnitude faster than competing kernel methods at test time. We also find that our method performs competitively on a completely different task, 3D shape classification. We perform these experiments to show that our method can be used as a fast, simple, and flexible baseline for rotation-invariant prediction problems on 3D point cloud data.

### 5.1 Small-Molecule Energy Regression

A common target for machine learning in chemistry is the prediction of a potential energy surface, which maps from 3D atomic configurations to the atomization energy of a molecule. Large standardized datasets such as QM7 (Blum & Reymond, 2009; Rupp et al., 2012) and QM9 (Ruddigkeit et al., 2012; Ramakrishnan et al., 2014) offer a convenient way to test the performance of these machine learning models across a wide chemical space of small organic molecules. Both datasets contain the 3D atomic coordinates at equilibrium and corresponding internal energies. The 3D coordinates and molecular properties are obtained by costly quantum mechanical calculations, so there may be settings which require high-throughput screening where an approximate but fast machine learning model may provide an advantageous alternative to classical methods. To adapt our method to this task, we make two small changes.

**Element-Type Encoding** The rotation-invariant random feature method defined above works for general, unlabeled point clouds. However, in the chemistry datasets mentioned above, we are given more information than just the 3D coordinates of particles in the molecule: the particles are individual atoms, and we know their element type. Incorporating the element type of individual atoms in a machine learning method is crucial for accurate prediction. Different elements interact in quantitatively different ways, as predicted by the laws of gravitation and electrostatic repulsion, and they interact in qualitatively different ways due to their electron configurations.

We use an element-type encoding method inspired by FCHL19 (Christensen et al., 2020) and ACE (Kovács et al., 2021). To construct this element-type encoding, we look at the local view of a molecule created by centering the atomic coordinates at a given atom, separate the atoms into different point clouds for each element type, and compute one random feature per element type. We then repeat this procedure for all elements of a given type and sum their feature vectors.

Expressed mathematically, we are given a sample $p_i = \{x_{i,1}, \ldots, x_{i,N_i}\}$ with charges $\{c_{i,1}, \ldots, c_{i,N_i}\}$. Let $p_i^{(c_j)}$ be the collection of atoms with charge $c_j$, and let $p_i^{(c_j)} - x_{i,h}$ denote the point cloud of atoms in $p_i$ with charge $c_j$ centered at point $x_{i,h}$. In this notation, $p_i^{(c_j)} - x_{i,h}$ specifies a point cloud, which we can treat as unlabeled because they all have the same charge. As before, a random feature is denoted $\varphi(\cdot; g_j)$. Then for all possible element-type pairs $(c_1, c_2)$, we compute individual entries in our feature matrix

$$\Phi_{i,j'} = \sum_{h:\ c_{i_h} = c_1} \varphi\left(p_i^{(c_2)} - x_{i,h}; g_j\right)$$

The column index $j'$ depends on $j, c_1$, and $c_2$.

**Radial Functions** We choose to parameterize our set of random functions as randomly-weighted spherical harmonics with fixed radial functions. The choice of radial functions appears as a design decision in many rotation-invariant machine learning methods. For example, radial functions appear as explicit design choices in Kovács et al. (2021); Christensen et al. (2020); Poulenard et al. (2019) and implicitly in the design of Cohen et al. (2018a). This design choice is often paramount to the empirical success of these methods. FCHL19 (Christensen et al., 2020) optimize their radial functions over multiple hyperparameters and carefully balance multiplicative terms including log-normal radial functions, polynomial decay, and soft cut-offs. The ACE models in Kovács et al. (2021) use a set of orthogonal polynomials defined over a carefully-chosen subset of

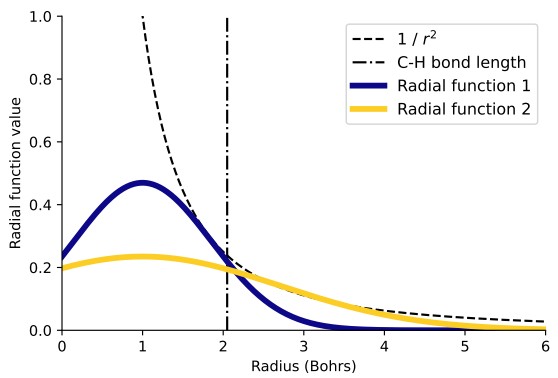

(a) Radial functions for small-molecule energy regression

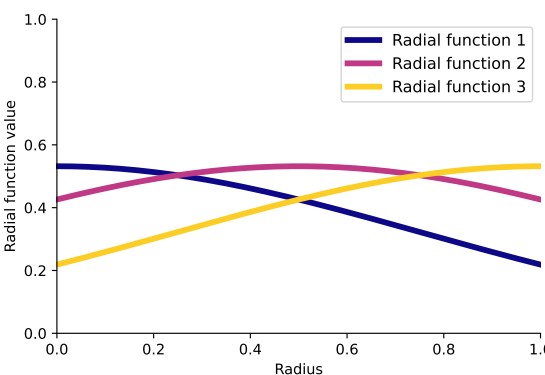

(b) Radial functions for 3D shape classification

Figure 2: Our rotation-invariant random feature method requires simple user-defined radial functions. Figure 2a shows the radial functions used in our small-molecule energy regression experiments. We include $\frac{1}{r^2}$ and the average Carbon-Hydrogen bond length to give context to the horizontal axis. Figure 2b shows the radial functions used in our 3D shape classification experiments. The 3D shapes in the benchmark dataset are normalized to fit inside the unit sphere.

the real line, a physically-motivated spatial transformation, and use extra ad-hoc radial functions that control the behavior of extremely nearby points. For our radial functions, we use two Gaussians, both centered at 1, with width parameters chosen so the full widths at half maximum are 2 and 4 respectively. Our radial functions are shown in Figure 2.

### 5.1.1 QM7 Atomization Energy Regression

The QM7 dataset contains 7,165 small molecules with element types H, C, N, O, S and a maximum of 7 heavy elements. We compare with two lightweight methods, FCHL19 and Spherical CNNs, that show learning results on the QM7 dataset. We reproduce their original training methods and compare test errors in Table 1. Notably, our rotation-invariant random feature method has average errors half of those of Spherical CNNs while being faster to train. Our best-performing model uses 2,000 random features. With the element-type encoding described above, this corresponds to 50,000 trainable parameters.

In this experiment, we use a training dataset of 5,732 samples and a test set of size 1,433. For our method and FCHL19, we use 90% of the training set to train the models and the remaining 10% as a validation set for hyperparameter optimization. To train the FCHL19 models, we use the validation set to optimize over Gaussian kernel widths and $L^2$ regularization parameters. In our method, we search over the number of random features and $L^2$ regularization parameters. We solve our ridge regression problem by taking a singular value decomposition of the random feature matrix, and then constructing a solution for each regularization level.

In addition to our ridge regression models, we also experiment using our random features as an input to deep neural networks. We are unable to find deep neural network models with reasonable prediction latencies that outperformed our ridge regression model (Appendix D). We also experimented with a model which combined a rotation-invariant PCA alignment (Xiao et al., 2020; Puny et al., 2022) with three-dimensional random features, but the alignments produced by this method were uninformative, and the model failed to meet basic baselines.

The space of small organic drug-like molecules is extremely large. The GDB-13 dataset (Blum & Reymond, 2009) contains nearly one billion reference molecules, even after filtering for molecule size and synthesis prospects. Most of the machine learning methods approximating potential energy surfaces are considered fast approximate replacements for costly quantum mechanical calculations. To screen the extremely large space of small organic drug-like molecules, these methods require high prediction throughput, which can

| Method | General-Purpose | Mean Absolute Error (eV) | Train Time (s) | Train Device |
|---|---|---|---|---|
| Spherical CNNs (Cohen et al., 2018a) | ✓ | 0.1565 | 546.7 | single GPU |
| Random Features (Ours) | ✓ | 0.0660 ± 0.00275 | 203.4 | 24 CPU cores |
| FCHL19 (Christensen et al., 2020) | ✗ | 0.0541 | 260.7 | 24 CPU cores |

Table 1: Test error and training time on the QM7 dataset. We report the mean absolute error on the test set for all methods and the standard error of the mean for our method. We discuss this experiment in Section 5.1.1.

be achieved by exploiting parallelism in the models' computational graphs and readily-available multicore CPU architectures. When parallelism inside the model is not available, the set of candidate molecules can be divided and data parallelism can be used to increase screening throughput. However, we consider another setting where experimental samples need to be analyzed in real time to make decisions about ongoing dynamic experiments. In this setting, prediction latency is paramount.

In the low-latency setting, our method can gracefully trade off prediction latency and prediction error by tuning the number of random features used in a given model. Importantly, all training samples are used to train the model. FCHL19 is a kernel method, so the time required to predict a new data point scales linearly with the number of training samples used. Thus, FCHL19 can only improve test latency at the cost of using fewer training samples and incurring higher test errors.

To explore the tradeoff between prediction latency and prediction error, we train random feature models and FCHL19 models of different sizes. We measure their prediction latencies and plot the results in Figure 3. We note that at similar error levels, FCHL19 models show prediction latencies that are an order of magnitude slower than ours. Spherical CNNs exhibit low test latency because their method is implemented for a GPU architecture, but their test errors are high, and the neural network approach does not admit obvious ways to achieve a tradeoff between latency and test error. The prediction latencies of FCHL19 and our random feature models depend on the number of atoms in the molecule, and we show this dependence in Figure 4.

### 5.1.2 QM9 Atomization Energy Regression

The QM9 dataset contains 133,885 molecules with up to 9 heavy atoms C, O, N, and F, as well as Hydrogen atoms. This is a larger and more complex dataset than QM7, and because there is more training data, neural network methods perform well on this benchmark. For this dataset, we follow Anderson et al. (2019); Gilmer et al. (2017) by constructing atomization energy as the difference between the internal energy at $0K$ and the thermochemical energy of a molecule's constituent atoms. We train a large-scale random feature model by taking 100,000 training samples from QM9 and generating 10,000 random features. We select an $L^2$ regularization parameter *a priori* by observing optimal regularization parameters from smaller scale experiments on a proper subset of the QM9 training data.

We compare the performance of our model with the reported errors of FCHL19 (Christensen et al., 2020) and other neural network methods. All methods considered in this comparison enforce rotational invariance in their predictions. The results of this comparison are shown in Table 2. Our rotation-invariant random feature method matches the performance of Cormorant, and it provides a very strong baseline against which we can quantify the effect of expert chemistry knowledge or neural network design. OrbNet (Qiao et al., 2020) uses a graph neural network architecture, and their inputs are carefully chosen from the results of density functional theory calculations in a rotation-invariant basis. PhysNet (Unke & Meuwly, 2019) uses an architecture heavily inspired by that of SCHNet (Schütt et al., 2018), and this iteration in model design resulted in almost a 50% reduction in test error.

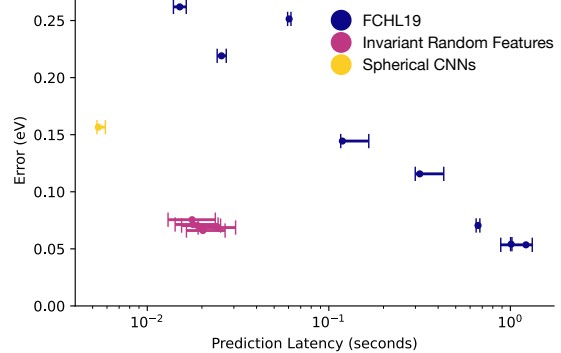

| Method | Model Size | Prediction Latency (s) | Train time (s) | Error (eV) |
|---|---|---|---|---|
| FCHL19 | 100 | 0.0151 | 0.4 | 0.2620 |
| FCHL19 | 200 | 0.0256 | 0.8 | 0.2191 |
| FCHL19 | 400 | 0.0604 | 2.3 | 0.2514 |
| FCHL19 | 800 | 0.1188 | 9.1 | 0.1444 |
| FCHL19 | 1600 | 0.3170 | 54.0 | 0.1157 |
| FCHL19 | 3200 | 0.6635 | 258.5 | 0.0705 |
| FCHL19 | 5000 | 1.0096 | 260.7 | 0.0541 |
| FCHL19 | 5732 | 1.2191 | 326.1 | 0.0535 |
| Random Features (Ours) | 250 | 0.0177 | 61.6 | 0.0755 |
| Random Features (Ours) | 500 | 0.0178 | 107.1 | 0.0714 |
| Random Features (Ours) | 750 | 0.0189 | 157.8 | 0.0704 |
| Random Features (Ours) | 1000 | 0.0203 | 203.4 | 0.0660 |
| Random Features (Ours) | 2000 | 0.0242 | 431.4 | 0.0687 |
| Spherical CNNs | N/A | 0.0054 | 546.7 | 0.1565 |

Figure 3: Invariant Random Features are faster at prediction time than FCHL19 and more accurate than Spherical CNNs. In this experiment, we compare the prediction latency and prediction error for models of different sizes evaluated on the QM7 dataset. For each method, we measure the latency of predicting each molecule on a held-out test set 5 times. In the figure, we plot the median prediction latencies with error bars spanning the $25^{th}$ and $75^{th}$ percentile measurements. The vertical axis of the figure is mean absolute error measured in eV on a held-out test set. The FCHL19 method is evaluated on 24 CPU cores using kernels with sample sizes 100, 200, 400, 800, 1600, 3200, 5000, and 5732. The rotation-invariant random features method is evaluated on 24 CPU cores with model sizes of 250, 500, 750, 1000, and 2000 random features. The Spherical CNNs method is evaluated on a single GPU. We discuss this experiment in Section 5.1.1.

| Method | General-Purpose | Model Type | Mean Absolute Error (eV) | Input Features |
|---|---|---|---|---|
| Cormorant (Anderson et al., 2019) | ✓ | General-purpose rotation-invariant architecture | 0.022 | • Charges
• Atomic locations |
| Random Features (Ours) | ✓ | Ridge Regression | 0.022 ± 4.45e-04 | • Charges
• Rotation-invariant random features |
| FCHL19 (Christensen et al., 2020) | ✗ | Kernel Ridge Regression | 0.011 | • Charges
• Interatomic distances
• 3-body angles |
| SchNet (Schütt et al., 2018) | ✗ | Neural Network with atomic interaction blocks | 0.014 | • Charges
• Interatomic distances |
| PhysNet (Unke & Meuwly, 2019) | ✗ | Neural Network with attention layers | 0.008 | • Charges
• Interatomic distances |
| OrbNet (Qiao et al., 2020) | ✗ | Message-Passing GNN | 0.005 | • Mean-field DFT calculations |

Table 2: Comparing large-scale models trained on QM9. FCHL19 (Christensen et al., 2020) was trained on 75,000 samples from QM9 and the other methods were trained on 100,000 samples. We report the mean absolute error on the test set for all methods and the standard error of the mean for our method. We discuss this experiment in Section 5.1.2.

Training models on large subsets of the QM9 dataset is difficult. FCHL19 requires 27 hours to construct a complete kernel matrix of size $(133{,}855 \times 133{,}855)$ on a compute node with 24 processors. The 27 hours does not include the time required to find the model's linear weights. The neural network methods require long training sequences on GPUs. Both Cormorant and PhysNet report training for 48 hours on a GPU.

When using a large sample size, our method is similarly difficult to train. We are able to construct a matrix of random features of size $(100{,}000 \times 250{,}000)$ in 27 minutes on a machine with 24 processors, but the matrix is highly ill-conditioned, and using an iterative method to approximately solve the ridge regression problem requires 76.5 hours. Using an iterative method introduces approximation error which we did not encounter when performing experiments on the other datasets. We attribute some of the performance gap between our method and FCHL19 to this approximation error. Fortunately, solving large, dense, overdetermined ridge regression problems is an active area of research, and it is quite likely that applying methods from this area of research to our problem would result in a speed-up. We outline some promising approaches in Appendix F. We believe the conditioning of our problem is caused by the choice of element-type encoding, and we leave finding a new element-type encoding that produces well-conditioned feature matrices for future work.

### 5.2 Shape Classification

To show that our method is general-purpose, we test our same method on a completely different task, 3D shape classification. In particular, we consider multiclass classification on the ModelNet40 benchmark dataset. The ModelNet40 benchmark dataset Zhirong Wu et al. (2015) is a set of computer-generated 3D models of common shapes, such as mugs, tables, and airplanes. We evaluate the performance of our model in three train/test settings, where the train and test sets either contain rotations about the $z$ axis, or arbitrary rotations in $SO(3)$. There are 9,843 training examples and 2,468 test examples spread across 40 different classes. The shape objects are specified by 3D triangular meshes, which define a (possibly disconnected) object surface. To generate a point cloud from individual objects in this dataset, one must choose a sampling strategy and sample points from the surface of the object. We use the dataset generated by Qi et al. (2017a), which samples 1,024 points on the mesh faces uniformly at random. The point clouds are then centered at the origin and scaled to fit inside the unit sphere.

We solve the multiclass classification problem with multinomial logistic regression. More specifically, we optimize the binary cross entropy loss with L2 regularization to learn a set of linear weights for each of the 40 classes in the dataset. At test time, we evaluate the 40 different linear models and predict by choosing the class with the highest prediction score. We also slightly change the definition of our data function; for a point cloud $p_i = \{x_{i,1}, ..., x_{i,N_i}\}$, we use a normalized data function $p_i(x) = \frac{1}{N_i} \sum_{j=1}^{N_i} \delta(x - x_{i,j})$ to eliminate any dependence on the number of points sampled from the surface of the shape objects. For this setting, we use radial functions that are three Gaussian bumps with centers at $0, 0.5$, and $1$, with width $\sigma = 0.75$.

We compare our method with another rotation-invariant method, SPHNet (Poulenard et al., 2019). SPHNet is a multilayer point-convolutional neural network that enforces rotational invariance by computing the spherical harmonic power spectrum of the input point cloud. SPHNet's rotation-invariant method also requires a choice of radial functions, and they use two Gaussians with different centers and vary the width of the Gaussians at different layers of their network. The results of our comparison are in Table 3. Our method does not achieve near state-of-the-art results, but we conclude it provides a strong baseline on this challenging multiclass problem. We discuss the prediction latency of our method on the ModelNet40 dataset in Appendix D.

## 6 Discussion and Future Work

Our method is a simple, flexible, and competitive baseline for rotationally-invariant prediction problems on 3D point clouds. Rotation-invariant random features are simple to explain and implement, and using them in varied prediction tasks requires a minimal amount of design choices. Our method does not achieve state-of-the-art accuracy on any prediction task, but it provides a competitive baseline that allows us to begin to quantify the effect of neural network models and expert chemistry knowledge in methods that enforce rotational invariance. In particular, we find that for molecular property prediction tasks, the inductive bias of rotation invariance is enough to provide very strong performance with our general-purpose random feature

| Method | Rotation Invariant | z/z | SO(3)/SO(3) | z/SO(3) |
|---|---|---|---|---|
| Spherical CNNs (Esteves et al., 2018) | ✓ | 0.889 | 0.869 | 0.786 |
| SPHNet (Poulenard et al., 2019) | ✓ | 0.789 | 0.786 | 0.779 |
| Vector Neurons (Deng et al., 2021) | ✓ | 0.902 | 0.895 | 0.895 |
| Random Features (Ours) | ✓ | 0.693 | 0.692 | 0.666 |
| PointNet++ (Qi et al., 2017b) | ✗ | 0.918 | 0.850 | 0.284 |

Table 3: Test accuracy on the ModelNet40 shape classification benchmark task. Our method sets a strong baseline to compare against neural network methods, especially in the challenging $z/SO(3)$ train/test regime. We discuss this experiment in Section 5.2.

model. For 3D shape classification, we find that the inductive bias of rotation invariance provides a non-trivial baseline, but the gap between our model and neural networks is larger.

Our method shows promise in settings where low prediction latency is desired. In high-throughput experiments, such as the ATLAS experiment at the CERN Large Hadron Collider (Collaboration, 2008), data is generated at such a high velocity that real-time decisions must be made whether to save or discard individual samples. For this type of initial screening task, we imagine our low-latency and flexible prediction method will be an attractive candidate. Another experimental use-case for low-latency models is as replacements for force fields in molecular dynamics simulations (Gilmer et al., 2017). In these simulations, the energy and forces in a molecular system is repeatedly queried in a serial fashion. Any improvement in the latency of energy or force prediction would translate to a speedup of the overall system. Because our model has an extremely shallow computational graph, we expect it will have much lower latency than deep neural networks once implemented on a GPU.

One can also interpret our work as an initial investigation into the use of random Fourier feature methods for approximating common kernel methods in computational chemistry. Kernel methods are well-studied and highly performant in machine learning for computational chemistry; examples of such methods include Rupp et al. (2012); Montavon et al. (2012); Bartók et al. (2013); Christensen et al. (2020). It is a natural question to ask whether these methods can benefit from low prediction latency while maintaining high test accuracy when approximated by random features. Our method does not implement an exact approximation to any of the kernel methods above, but our experiments show promise in this area. Our test errors are near those of FCHL19 on the QM7 dataset, and versions of our model using only 10% of the optimal number of parameters have reasonable test errors. However, when training our model on the QM9 dataset, we have seen that the particular element-type encoding we have used creates ill-conditioned feature matrices and makes optimization difficult. To fully realize the potential of random feature methods in accelerating kernel learning for computational chemistry, a new element-type encoding is needed.

## 7 Broader Impact Statement

The task of molecular property prediction is important to a wide range of applications, including the development of new pharmaceuticals, materials, and solvents. Some of these applications may be misused.

## Acknowledgments

RMW was supported by AFSOR award FA9550-18-1-0166 and NSF DMS-2023109. OJM was supported by an NSF Research Traineeship under grant NSF 2022023.

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

## A  Full Integration Details

We wish to solve the following integral, which appears in the definition of rotation-invariant random features Equation (6):

$$\int_{SO(3)} \langle Q \circ p_i, g \rangle^2 dQ \tag{11}$$

From Section 4.1, we know this integral can decompose into

$$\int_{SO(3)} \langle Q \circ p_i, g \rangle^2 dQ = \sum_{j_1,j_2=1}^{N_i} \int_{SO(3)} g(Qx_{i,j_1})g(Qx_{i,j_2})dQ \tag{12}$$

And we also have chosen a particular functional form for our random function $g$:

$$g(x) = \sum_{k=1}^{K} \sum_{\ell=0}^{L} \sum_{m=-\ell}^{\ell} w_{m,k}^{(\ell)} Y_m^{(\ell)}(\hat{x}) R_k(\|x\|) \tag{13}$$

Repeated application of the linearity of integration gives us:

$$\int_{SO(3)} \langle Q \circ p_i, g \rangle^2 dQ = \tag{14}$$

$$= \sum_{j_1,j_2=1}^{N} \int_{SO(3)} g(Qx_{i,j_1})g(Qx_{i,j_2})dQ \tag{15}$$

$$= \sum_{j_1,j_2=1}^{N_i} \sum_{k_1,k_2=1}^{K} \sum_{\ell_1,\ell_2=0}^{L} \sum_{m_1=-\ell_1}^{\ell_1} \sum_{m_2=-\ell_2}^{\ell_2} w_{m_1,k_1}^{(\ell_1)} w_{m_2,k_2}^{(\ell_2)} R_{k_1}(\|x_{i,j_1}\|) R_{k_2}(\|x_{i,j_2}\|)$$

$$\int_{SO(3)} Y_{m_1}^{(\ell_1)}(Q\hat{x}_{i,j_1}) Y_{m_2}^{(\ell_2)}(Q\hat{x}_{i,j_2}) dQ \quad (16)$$

We can now focus on the integral in Equation (16), apply the rotation rule for spherical harmonics introduced in Section 3.1, and again exploit the linearity of the integral.

$$\int_{SO(3)} Y_{m_1}^{(\ell_1)}(Q\hat{x}_{i,j_1}) Y_{m_2}^{(\ell_2)}(Q\hat{x}_{i,j_2}) dQ \quad (17)$$

$$= \int_{SO(3)} \left( \sum_{m_1'=-\ell_1}^{\ell_1} Y_{m_1'}^{(\ell_1)}(\hat{x}_{i,j_1}) D^{(\ell_1)}(Q)_{m_1,m_1'} \right) \left( \sum_{m_2'=-\ell_2}^{\ell_2} Y_{m_2'}^{(\ell_2)}(\hat{x}_{i,j_2}) D^{(\ell_2)}(Q)_{m_2,m_2'} \right) dQ \quad (18)$$

$$= \sum_{m_1'=-\ell_1}^{\ell_1} \sum_{m_2'=-\ell_2}^{\ell_2} Y_{m_1'}^{(\ell_1)}(\hat{x}_{i,j_1}) Y_{m_2'}^{(\ell_2)}(\hat{x}_{i,j_2}) \int_{SO(3)} D^{(\ell_1)}(Q)_{m_1,m_1'} D^{(\ell_2)}(Q)_{m_2,m_2'} dQ \quad (19)$$

$$\quad (20)$$

At this point, we are able to apply the integration rule for elements of Wigner-D matrices introduced in Section 3.1.

$$\int_{SO(3)} Y_{m_1}^{(\ell_1)}(Q\hat{x}_{i,j_1}) Y_{m_2}^{(\ell_2)}(Q\hat{x}_{i,j_2}) dQ \quad (21)$$

$$= \sum_{m_1'=-\ell_1}^{\ell_1} \sum_{m_2'=-\ell_2}^{\ell_2} Y_{m_1'}^{(\ell_1)}(\hat{x}_{i,j_1}) Y_{m_2'}^{(\ell_2)}(\hat{x}_{i,j_2}) (-1)^{m_1-m_1'} \frac{8\pi^2}{2\ell+1} \delta_{-m_1,m_2} \delta_{-m_1',m_2'} \delta_{\ell_1,\ell_2} \quad (22)$$

$$= \delta_{\ell_1,\ell_2} \delta_{-m_1,m_2} \frac{8\pi^2}{2\ell+1} \sum_{m_1'=-\ell_1}^{\ell_1} (-1)^{m_1-m_1'} Y_{m_1'}^{(\ell_1)}(\hat{x}_{i,j_1}) Y_{-m_1'}^{(\ell_1)}(\hat{x}_{i,j_2}) \quad (23)$$

To put it all together, we substitute Equation (23) into Equation (16), and we see many terms drop out of the sums:

$$(16) = \sum_{j_1,j_2=1}^{N_i} \sum_{k_1,k_2=0}^{K} \sum_{\ell_1,\ell_2=0}^{L} \sum_{m_1=-\ell_1}^{\ell_1} \sum_{m_2=-\ell_2}^{\ell_2} w_{m_1,k_1}^{(\ell_1)} w_{m_2,k_2}^{(\ell_2)} R_{k_1}(\|x_{i,j_1}\|) R_{k_2}(\|x_{i,j_2}\|)$$

$$\delta_{\ell_1,\ell_2} \delta_{-m_1,m_2} \frac{8\pi^2}{2\ell+1} \sum_{m_1'=-\ell_1}^{\ell_1} (-1)^{m_1-m_1'} Y_{m_1'}^{(\ell_1)}(\hat{x}_{i,j_1}) Y_{-m_1'}^{(\ell_1)}(\hat{x}_{i,j_2}) \quad (24)$$

$$\int_{SO(3)} \langle Q \circ p_i, g \rangle^2 dQ = \sum_{j_1,j_2=1}^{N_i} \sum_{k_1,k_2=0}^{K} \sum_{\ell_1=0}^{L} \sum_{m_1=-\ell_1}^{\ell_1} w_{m_1,k_1}^{(\ell_1)} w_{-m_1,k_2}^{(\ell_1)} R_{k_1}(\|x_{i,j_1}\|) R_{k_2}(\|x_{i,j_2}\|)$$

$$\frac{8\pi^2}{2\ell+1} \sum_{m_1'=-\ell_1}^{\ell_1} (-1)^{m_1-m_1'} Y_{m_1'}^{(\ell_1)}(\hat{x}_{i,j_1}) Y_{-m_1'}^{(\ell_1)}(\hat{x}_{i,j_2}) \quad (25)$$

Once the spherical harmonics and radial functions are evaluated, this sum has $O(N^2 K^2 L^3)$ terms.

## A.1 Basis Expansion

When implementing Equation (25) to compute rotationally-invariant random features, one can see that a large amount of computational effort can be re-used between different random features evaluated on the same

sample. A table of spherical harmonic evaluations for all the points $\{\hat{x}_{i,1}, \ldots \hat{x}_{i,N_i}\}$ can be pre-computed once and re-used. Also, by re-arranging the order of summation in Equation (25), we arrive at this expression:

$$\int_{SO(3)} \langle Q \circ p_i, g \rangle^2 dQ = \sum_{k_1,k_2=1}^{K} \sum_{\ell_1=0}^{L} \sum_{m_1=-\ell_1}^{\ell_1} w_{m_1,k_1}^{(\ell_1)} w_{-m_1,k_2}^{(\ell_1)}$$

$$\sum_{j_1,j_2=1}^{N_i} R_{k_1}(\|x_{i,j_1}\|) R_{k_2}(\|x_{i,j_2}\|) \frac{8\pi^2}{2\ell+1} \sum_{m_1'=-\ell_1}^{\ell_1} (-1)^{m_1-m_1'} Y_{m_1'}^{(\ell_1)}(\hat{x}_{i,j_1}) Y_{-m_1'}^{(\ell_1)}(\hat{x}_{i,j_2}) \quad (26)$$

Pre-computing everything after the line break allows for the re-use of a large amount of computational work between different random feature evaluations. We can interpret this pre-computation as an expansion of our point cloud in a rotationally-invariant basis $B$, which depends on the choice of radial function. Appendix B relates this basis expansion to the Atomic Cluster Expansion method.

$$B[\ell, m, k_1, k_2] = \sum_{j_1,j_2=1}^{N_i} R_{k_1}(\|x_{i,j_1}\|) R_{k_2}(\|x_{i,j_2}\|) \frac{8\pi^2}{2\ell+1} \sum_{m'=-\ell}^{\ell} (-1)^{m-m'} Y_{m'}^{(\ell)}(\hat{x}_{i,j_1}) Y_{-m'}^{(\ell)}(\hat{x}_{i,j_2}) \quad (27)$$

$$= \sum_{j_1,j_2=1}^{N_i} R_{k_1}(\|x_{i,j_1}\|) R_{k_2}(\|x_{i,j_2}\|) \int_{SO(3)} Y_m^{(\ell)}(Q\hat{x}_{i,j_1}) Y_{-m}^{(\ell)}(Q\hat{x}_{i,j_2}) dQ \quad (28)$$

In our implementation we precompute this $B$ tensor once for each sample, and then we perform a tensor contraction with the random weights:

$$\int_{SO(3)} \langle Q \circ p_i, g \rangle^2 dQ = \sum_{k_1,k_2=1}^{K} \sum_{\ell_1=0}^{L} \sum_{m_1=-\ell_1}^{\ell_1} w_{m_1,k_1}^{(\ell_1)} w_{-m_1,k_2}^{(\ell_1)} B[\ell_1, m_1, k_1, k_2] \quad (29)$$

# B  Connection Between Our Method and Randomized ACE Methods

## B.1  Overview of the ACE basis

The atomic cluster expansion (ACE) method builds a rotationally-invariant basis for point clouds that are an extension of our pre-computed basis expansion described in Appendix A.1. In the ACE method, a preliminary basis $A_{k,\ell,m}$ is formed by projecting a point cloud function $p_i(x) = \sum_j \delta(x - x_{i_j})$ onto a single-particle basis function $\varphi_{k,\ell,m}(x)$:

$$\varphi_{k,\ell,m}(x) = R_k(\|x\|) Y_m^{(\ell)}(x) \quad (30)$$

$$A_{k,\ell,m}(p_i) = \langle \varphi_{k,\ell,m}, p_i \rangle \quad (31)$$

$$= \sum_j \varphi_{k,\ell,m}(x_{i_j}) \quad (32)$$

where $R_k$ is a set of radial functions. In Kovács et al. (2021), this set of radial functions is defined by taking a nonlinear radial transformation, and then the radial functions are a set of orthogonal polynomials defined on the transformed radii. This basis set is augmented with auxiliary basis functions to ensure the potential energy of two atoms at extremely nearby points diverges to infinity.

The $A_{k,\ell,m}$ basis is a first step, but it only models single particles at a time. To model pairwise particle interactions, or higher-order interactions, the $A$ basis is extended by taking tensor products. First, a "correlation order" $N$ is chosen. Then, the $A$ basis is extended via tensor products:

$$A_{\underline{k},\underline{\ell},\underline{m}}(p_i) = \prod_{\alpha=1}^{N} A_{k_\alpha,\ell_\alpha,m_\alpha}(p_i) \quad (33)$$

where $\underline{k} = (k_\alpha)_{\alpha=1}^N$, $\underline{\ell} = (\ell_\alpha)_{\alpha=1}^N$, and $\underline{m} = (m_\alpha)_{\alpha=1}^N$ are multi-indices.

Finally, to ensure invariance with respect to permutations, reflections, and rotations of the point cloud, the ACE method forms a Haar measure $dg$ over $O(3)$ and computes a symmetrized version of the $A$ basis:

$$B_{\underline{k},\underline{\ell},\underline{m}}(p_i) = \int_{O(3)} A_{\underline{k},\underline{\ell},\underline{m}}(H \circ p_i)dH \tag{34}$$

Drautz (2019) gives a detailed description of how this integral is performed. Applications of representation theory give concise descriptions of which multi-indices $\underline{k}, \underline{\ell}, \underline{m}$ remain after performing this integral; many are identically zero.

## B.2 Connection to Our Method

If one chooses to compute an ACE basis with correlation order $N = 2$, the resulting basis is very similar to our pre-computed basis expansion in Equation (28).

$$B_{\underline{k},\underline{\ell},\underline{m}}(p_i) = \int_{O(3)} A_{\underline{k},\underline{\ell},\underline{m}}(H \circ p_i)dH \tag{35}$$

$$= \int_{O(3)} A_{k_1,\ell_1,m_1}(H \circ p_i)A_{k_2,\ell_2,m_2}(H \circ p_i)dH \tag{36}$$

$$= \int_{O(3)} \left( \sum_{j_1} \varphi_{k_1,\ell_1,m_1}(Hx_{i_{j_1}}) \right) \left( \sum_{j_2} \varphi_{k_2,\ell_2,m_2}(Hx_{i_{j_2}}) \right) dH \tag{37}$$

$$= \int_{O(3)} \left( \sum_{j_1} R_{k_1}(\|x_{i_{j_1}}\|)Y_{m_1}^{(\ell_1)}(H\hat{x}_{i_{j_1}}) \right) \left( \sum_{j_2} R_{k_2}(\|x_{i_{j_2}}\|)Y_{m_2}^{(\ell_2)}(H\hat{x}_{i_{j_2}}) \right) dH \tag{38}$$

$$= \sum_{j_1,j_2} R_{k_1}(\|x_{i_{j_1}}\|)R_{k_2}(\|x_{i_{j_2}}\|) \int_{O(3)} Y_{m_1}^{(\ell_1)}(H\hat{x}_{i_{j_1}})Y_{m_2}^{(\ell_2)}(H\hat{x}_{i_{j_2}})dH \tag{39}$$

The remaining difference between our basis expansion in Equation (28) and the ACE basis expansion with correlation order $N = 2$ is the ACE basis integrates over all of $O(3)$, while our method only enforces invariance with respect to $SO(3) \subset O(3)$.

So one can describe our random features as random nonlinear projections of a simplified version of the ACE basis with correlation order $N = 2$ using extremely simple radial functions.

## B.3 Benefit of Random Features

We conduct the following experiment to quantify the benefit of using random nonlinear features over fitting a linear model in our simplified version of the ACE basis. Using the same hyperparameters as in our energy regression experiments, we compute our basis expansion for all the samples in the QM7 dataset. Using the same train/validation/test split, we perform ridge regression by fitting a linear model in our basis $B$ with $L^2$ regularization. We use the validation set to identify the best-performing model and report errors on the test set in Table 4. We note that the best-performing linear model has more than twice the error of the best performing random features model.

# C  Representation Theory of the 3D Rotation Group

## C.1  Conjugation of Wigner-D Matrices

In the following, we derive a formula for conjugating an element of a Wigner-D matrix. First, we use definition 6.44 in Thompson (1994) for the Wigner-D matrices:

$$D_{m',m}^{(\ell)}(\alpha, \beta, \gamma) = e^{-im'\alpha}d_{m',m}^{(\ell)}(\beta)e^{-im\gamma} \tag{40}$$

| Method | Test Error (eV) |
|---|---|
| Spherical CNNs | 0.1565 |
| FCHL19 | 0.0541 |
| Random Features (Ours) | 0.0660 |
| Linear Model of $B$ basis (Ours) | 0.1542 |

Table 4: Reported are test errors on the QM7 dataset for Spherical CNNs, FCHL19, and two variants of our method. The first variant is the full rotation-invariant random features model. The second variant is a linear model of our basis expansion. We note that while our random feature model is the second-best performing model on this benchmark, our linear model is the nearly the worst-performing model.

Here, $(\alpha, \beta, \gamma)$ are Euler angles parameterizing a rotation, and $d^{(\ell)}_{m',m}$ is an element of a Wigner-d (small d) matrix. We omit the exact definition of $d^{(\ell)}_{m',m}$, but we note the following symmetry properties:

$$d^{(\ell)}_{-m',-m}(\beta) = d^{(\ell)}_{m',m}(-\beta) \tag{41}$$

$$d^{(\ell)}_{m',m}(-\beta) = d^{(\ell)}_{m,m'}(\beta) \tag{42}$$

$$d^{(\ell)}_{m,m'}(\beta) = (-1)^{m-m'} d^{(\ell)}_{m',m}(\beta) \tag{43}$$

The relationship between the Wigner d matrix elements allows us to derive a formula for conjugating an element of a Wigner D matrix:

$$D^{(\ell)}_{m',m}(\alpha, \beta, \gamma)^* = e^{im'\alpha} d^{(\ell)}_{m',m}(\beta)^* e^{im\gamma} \tag{44}$$

$$= e^{-i(-m')\alpha} d^{(\ell)}_{m',m}(\beta) e^{-i(-m)\gamma} \tag{45}$$

$$= e^{-i(-m')\alpha} d^{(\ell)}_{-m',-m}(-\beta) e^{-i(-m)\gamma} \tag{46}$$

$$= e^{-i(-m')\alpha} d^{(\ell)}_{-m,-m'}(\beta) e^{-i(-m)\gamma} \tag{47}$$

$$= e^{-i(-m')\alpha} (-1)^{-m+m'} d^{(\ell)}_{-m',-m}(\beta) e^{-i(-m)\gamma} \tag{48}$$

$$= (-1)^{m'-m} D^{(\ell)}_{-m',-m}(\alpha, \beta, \gamma) \tag{49}$$

## C.2 Orthogonality Relations

We use the following orthogonality relationship between elements of Wigner D matrices:

$$\int_{SO(3)} D^{(\ell_1)}(Q)^*_{m_1,k_1} D^{(\ell_2)}(Q)_{m_2,k_2} dQ = \frac{8\pi^2}{2\ell+1} \delta_{m_1,m_2} \delta_{k_1,k_2} \delta_{\ell_1,\ell_2} \tag{50}$$

This is a well-known fact about the Wigner-D matrices. For a proof, one can derive this relationship from Schur orthogonality relations, which are a consequence of Schur's lemma. A direct calculation can be found in section 6.4.2 of Thompson (1994). Combining the two facts from above, we arrive at the following identity, which we use to compute our rotation-invariant random features.

$$\int_{SO(3)} D^{(\ell_1)}(Q)_{m_1,k_1} D^{(\ell_2)}(Q)_{m_2,k_2} dQ = (-1)^{m_1-k_1} \frac{8\pi^2}{2\ell+1} \delta_{-m_1,m_2} \delta_{-k_1,k_2} \delta_{\ell_1,\ell_2} \tag{51}$$

# D Additional Latency Experiments

In this section we present extra experiments about the prediction latency of our method. Figure 4 compares the effect of molecule size on prediction latency for our method and FCHL19. Importantly, both methods have

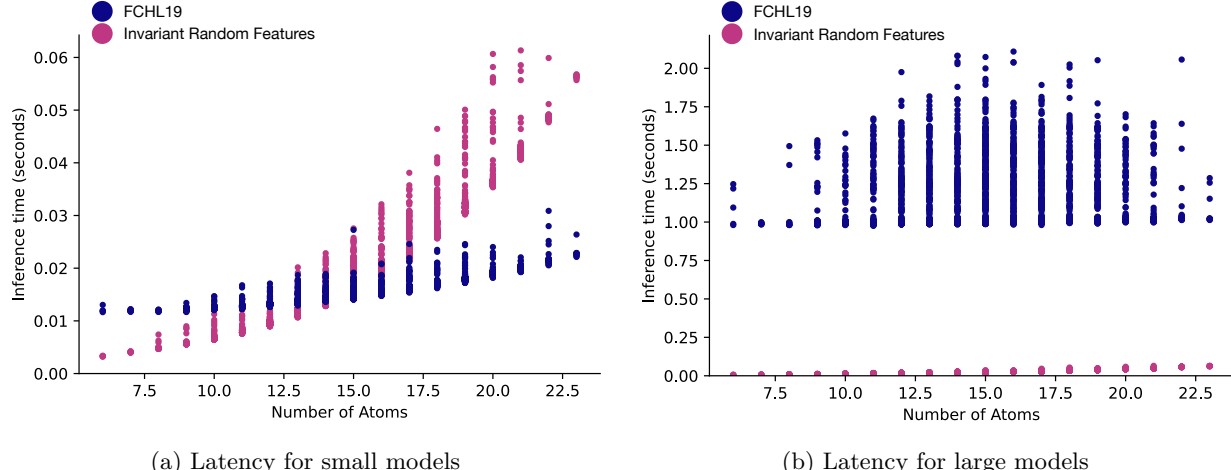

(a) Latency for small models  (b) Latency for large models

Figure 4: The number of atoms in a molecule affects the prediction latency for both FCHL19 and the invariant random features method. In Figure 4a, we show the prediction latencies for the smallest models tested, corresponding to 250 random features for our method and 200 training samples for FCHL19. For this model size, a quadratic scaling in the number of atoms largely determines the prediction latency. Figure 4b shows that for larger models (2,000 random features for our method and 5,000 training samples for FCHL19), the random features method is still in a regime where prediction latency is determined by the number of atoms, while FCHL19's prediction latency is dominated by the number of inner products required to evaluate the kernel.

a precomputation step that has complexity $O(N^2)$ where $N$ is the number of atoms. For our experiments on ModelNet40, the quadratic dependence on the number of points causes slow prediction latency, but we are able to alleviate this latency by reducing other hyperparameters (Figure 5).

We also investigate whether using deep multilayer perceptrons (MLPs) provides an advantageous accuracy-latency tradeoff compared to our original ridge regression method. We first investigate the prediction latency incurred by using MLPs with different widths and depths. In Figure 6, we observe that models with widths 512 and 1024 only moderately increase prediction latency and are therefore good architecture candidates for the prediction step. Models with higher widths incur up to four times the prediction latency of ridge regression. However, in Figure 7, we find that after training these architectures to convergence, their test error does not outperform ridge regression. The input to our MLPs were 2,000 random features from the QM7 dataset. We trained the MLPs to predict atomization energy by minimizing the mean squared error using the Adam optimizer. We searched over learning rate schedules and weight decay hyperparameters.

## E   Hyperparameter Sensitivity Analysis

We perform multiple experiments to evaluate the sensitivity of our random feature models' performance. In Figures 8 and 9, we find that our models trained on the QM7 dataset are most sensitive to $\sigma$, the standard deviation of the random weights. In Figures 10 and 11, we find the same result for models trained on the ModelNet40 dataset.

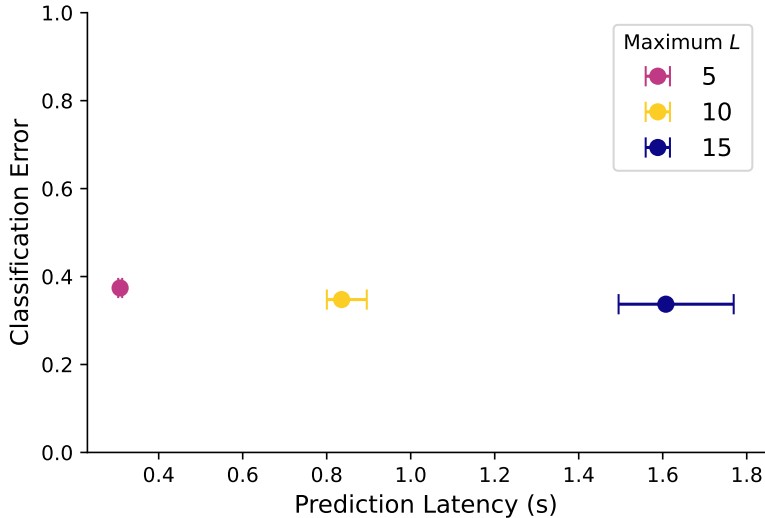

Figure 5: When training classification models on the ModelNet40 dataset, we are able to trade off prediction accuracy for prediction latency by changing the maximum frequency of spherical harmonics $L$. We represent samples from the ModelNet40 dataset using point clouds of size $N$=1,024. Computing random features has computational complexity scaling quadratically in $N$, which significantly slows down prediction latency. However, the prediction error of our model is relatively insensitive to the maximum frequency $L$ (see Figure 10a), so we can greatly reduce the prediction latency by decreasing $L$. We evaluate prediction latency on a machine with 24 physical CPU cores. In the figure, the centers show the median latency sample, and the error bars span the $25^{th}$ and $75^{th}$ percentiles.

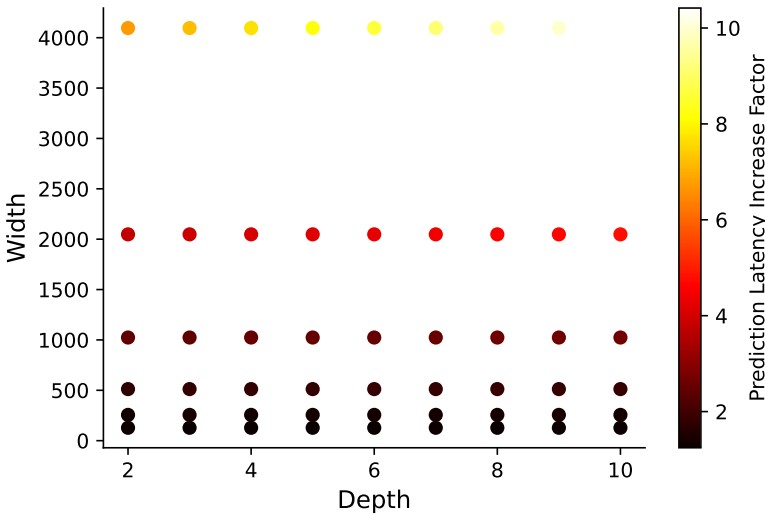

Figure 6: We measure the prediction latency of computing 2,000 random features and predicting using different MLP architectures. We present increase in latency when compared to ridge regression. A value of 2.0 indicates that predictions take twice as long as predictions using ridge regression.

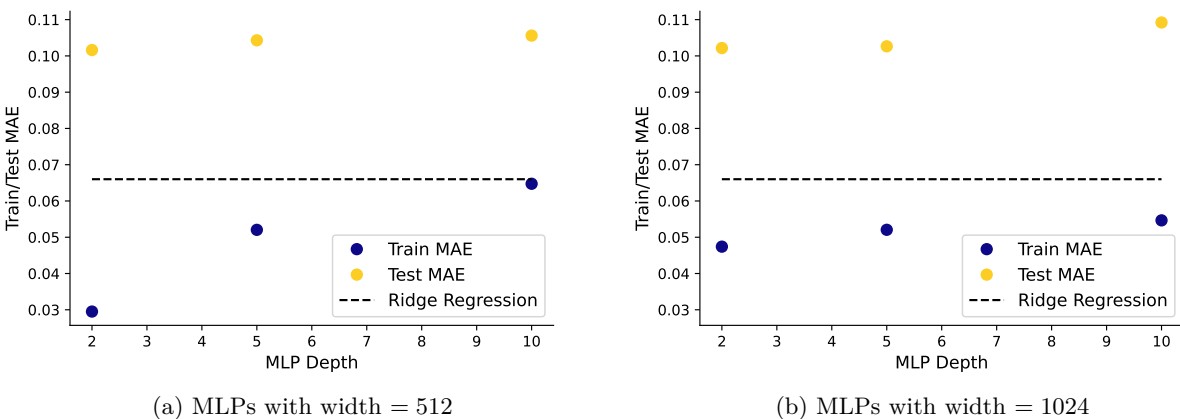

(a) MLPs with width $= 512$     (b) MLPs with width $= 1024$

Figure 7: The ridge regression baseline outperforms MLP architectures with low prediction latency.

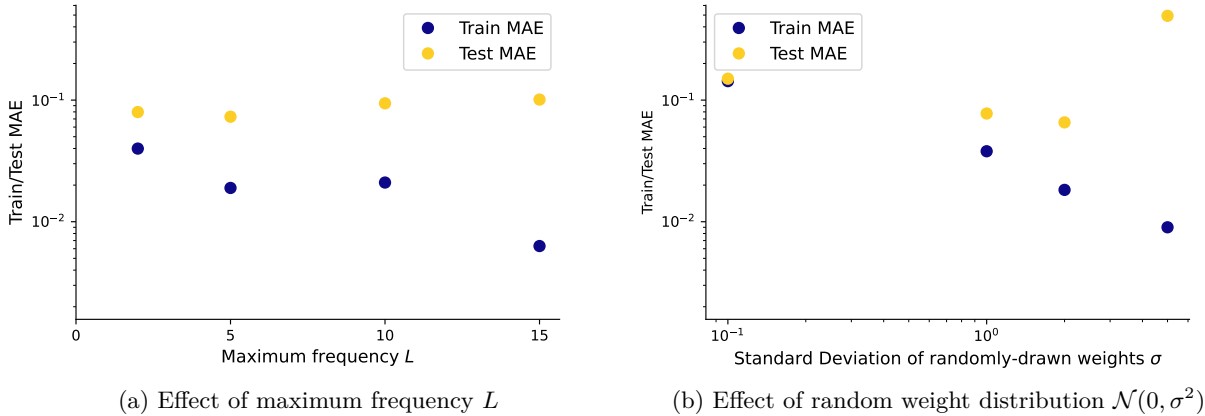

(a) Effect of maximum frequency $L$     (b) Effect of random weight distribution $\mathcal{N}(0, \sigma^2)$

Figure 8: In the QM7 experiments, the standard deviation of the weights is the most important hyperparameter. For this sensitivity experiment, we use a standard hyperparameter setting and vary one hyperparameter at a time. The standard setting is 2,000 random features; standard deviation of the random weights $\sigma = 2.0$; and the maximum frequency $L = 5$. In all experiments, we search over a pre-defined grid of $L^2$ regularization parameters, and select the best model using a held-out validation set. In Figure 8a, we vary the maximum frequency of the spherical harmonics, and we see this does not have a large effect on the test error. In Figure 8b, we vary $\sigma$, and we see this has a large effect on both the train and test error.

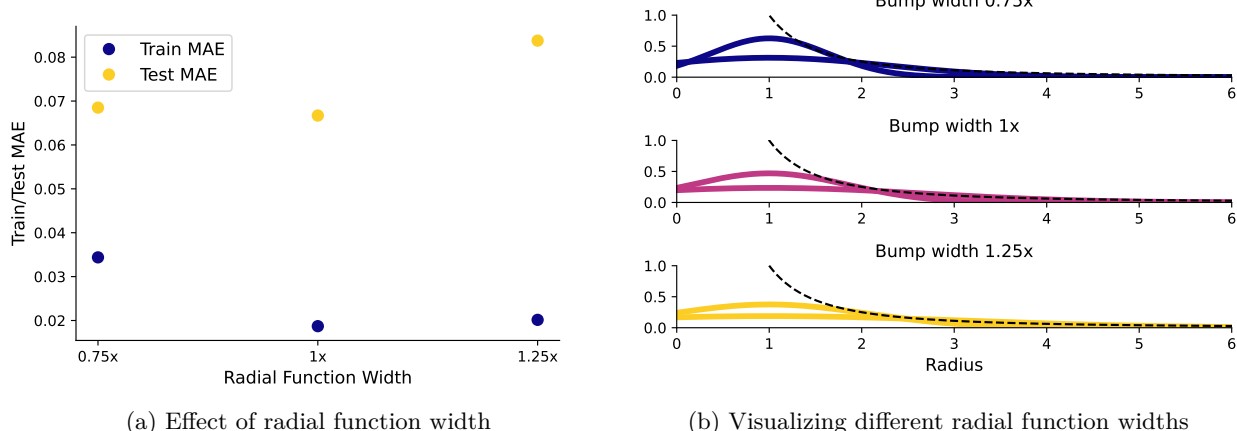

(a) Effect of radial function width

(b) Visualizing different radial function widths

Figure 9: Our results on the QM7 dataset are slightly sensitive to the width of the chosen radial functions. For the molecular energy regression experiments, our radial functions are two Gaussians centered at 1, with different widths $[\sigma_1, \sigma_2]$. We train and test models with different width scales: $0.75 \times [\sigma_1, \sigma_2]$ and $1.25 \times [\sigma_1, \sigma_2]$. We show the resulting train and test errors in Figure 9a and visualize the resulting radial functions in Figure 9b.

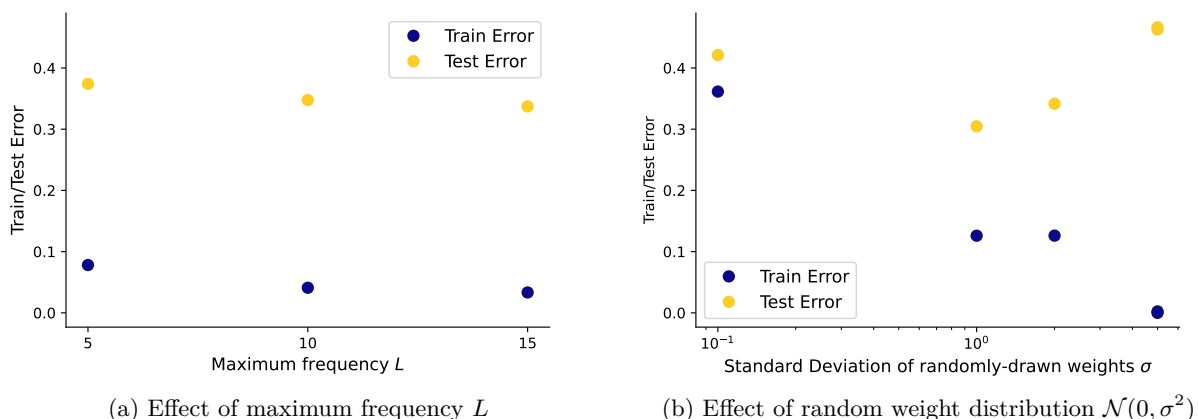

(a) Effect of maximum frequency $L$

(b) Effect of random weight distribution $\mathcal{N}(0, \sigma^2)$

Figure 10: In the ModelNet40 experiments, the standard deviation of the weights is the most important hyperparameter. For this sensitivity experiment, we use a standard hyperparameter setting and vary one hyperparameter at a time. The standard setting is 15,000 random features; standard deviation of the random weights $\sigma = 2.0$; and the maximum frequency $L = 10$. In Figure 10a, we vary the maximum frequency of spherical harmonics $L$, and we see that increasing $L$ slightly improves the performance. In Figure 10b, we vary $\sigma$, and we see the model's performance is sensitive to this parameter.

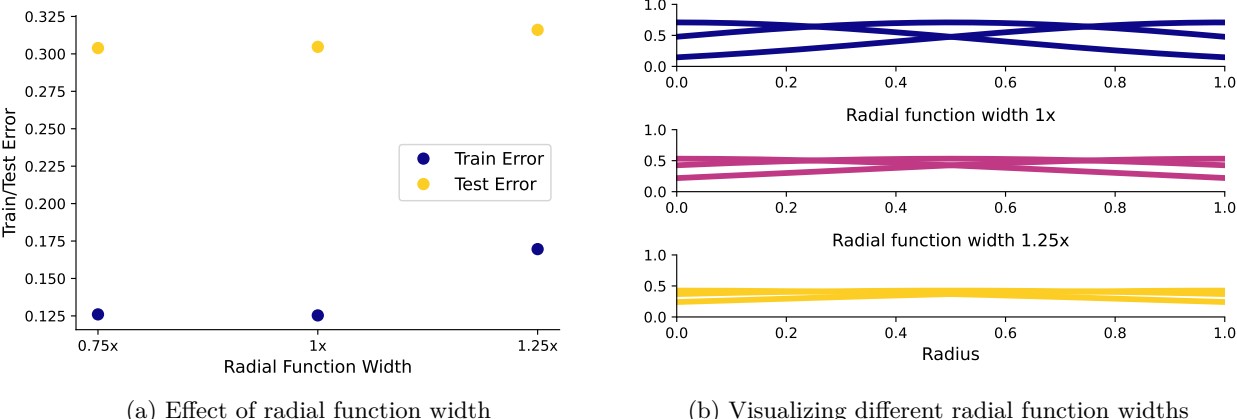

(a) Effect of radial function width

(b) Visualizing different radial function widths

Figure 11: Our results on the ModelNet40 dataset are very slightly sensitive to the width of the chosen radial functions. Our original radial functions are three Gaussians with centers $[0, 0.5, 1.0]$ and width $\sigma$. We train and test models with different radial function widths: $0.75 \times \sigma$, $1 \times \sigma$, and $1.25 \times \sigma$. Figure 11a shows the resulting train and test errors, and Figure 11b shows the resulting radial functions.

# F Solving Ridge Regression Problems for Random Features

This section outlines numerical methods for solving ridge regression problems arising in our random features models. In general, we want to solve the following problem:

$$\arg\min_{\beta} \|\Phi\beta - y\|_2^2 + \lambda\|\beta\|_2^2 \tag{52}$$

In our problem instances, $\Phi \in \mathbb{R}^{n \times d}$ is the matrix of random features. $\Phi$ is a dense matrix; there is no structured sparsity. $n$ is fixed; it is the number of samples in the dataset. The number of random features $d$ can be treated as a hyperparameter, but we find that $d > n$ is required for optimal test error, so the system of equations is underdetermined. $\lambda$ is the regularization parameter. Larger $\lambda$ corresponds to more regularization. Generally, when solving a large system, we approximately know a good value for $\lambda$ and want to keep it fixed.[2] To recap, we have an instance of a dense, overdetermined ridge regression problem.

By taking the gradient of Equation (52) with respect to $\beta$ we arrive at the normal equation:

$$0 = \left(\Phi^\top\Phi + \lambda I\right)\beta^* - \Phi^\top y \tag{53}$$

## F.1 Exact Solution via the Singular Value Decomposition

From Equation (53), we see that the solution to the ridge regression problem is

$$\begin{aligned} \beta^* &= \left(\Phi^\top\Phi + \lambda I\right)^{-1}\Phi^\top y \\ &= V\left(\Sigma^2 + \lambda I\right)^{-1}\Sigma^\top U^\top y \end{aligned} \tag{54}$$

where $\Phi = U\Sigma V^\top$ is the singular value decomposition (SVD). Because $\left(\Sigma^2 + \lambda I\right)$ is a diagonal matrix, we can exactly and efficiently compute its inverse. This suggests a method for solving Equation (52): compute the SVD of $\Phi$, efficiently compute $\left(\Sigma^2 + \lambda I\right)^{-1}$, and reconstruct $\beta^*$ via Equation (54). This solution method is stable and relatively performant for small problem instances ($n, d < 10,000$). We use this method in the QM7 experiments. Evaluating the full SVD of the feature matrix generated in the QM9 experiment is prohibitively costly, so we experimented with other solution methods.

## F.2 Approximate Solution via Iterative Methods

Equation (52) can be treated as an optimization problem in the variables $\beta \in \mathbb{R}^d$. The objective is strongly convex, with smoothness $\sigma_{\max}^2(\Phi)$ and strong convexity parameter[3] $\lambda$. This motivates the application of standard iterative methods to find an approximate solution. Gradient descent, for instance, is known to enjoy linear convergence (on a log-log plot) for smooth and strongly-convex objectives. However, the rate of convergence is $\frac{\lambda}{\sigma_{\max}^2(\Phi)}$, which is very slow for typical problem instances.

A family of iterative algorithms known as Krylov subspace methods are designed for linear least-squares problems like Equation (52). These methods include the conjugate gradient method and variations. In the QM9 experiment, we use LSQR (Paige & Saunders, 1982a;b) implemented in scipy (Virtanen et al., 2020).

While they are empirically faster than applying gradient descent to the objective and are known to converge to the exact solution in a finite number of iterations, the conjugate gradient method and LSQR both suffer from poor convergence dependence on the quantity $\frac{\sigma_{\max}^2(\Phi)}{\lambda}$. We have found that on our problem instance, LSQR requires approximately 100,000 iterations to converge to a solution with low training error.

## F.3 Approximate Solution via Principal Components Regression

We also attempt to approximately solve the ridge regression problem with principal components regression. We compute a truncated SVD with scipy's sparse linear algebra package (Virtanen et al., 2020). The runtime

---

[2]As opposed to other optimization problems, where increasing $\lambda$ is an acceptable method of improving the conditioning of the system.

[3]The strong convexity parameter is actually $\max\{\lambda, \sigma_{\min}^2(\Phi)\}$, but in practice, our random feature matrices are approximately low rank, so $\sigma_{\min}^2(\Phi) \ll \lambda$.

of this operation depends on $k$, the number of singular vectors we choose to resolve. After computing the truncated SVD, we construct an approximate solution:

$$\hat{\beta}_k = V_k \left( \Sigma_k^2 + \lambda I \right)^{-1} \Sigma_k^\top U_k^\top y$$

Empirically, we find that we need to choose a large $k$ to find a solution with low training error, which incurs a large runtime. We find LSQR arrives at an approximate solution faster than principal components regression.

### F.4 Future Work

We have identified two different classes of methods from numerical linear algebra as promising candidates for improving our scaling to larger datasets.

#### F.4.1 Preconditioning Iterative Methods

The runtime of iterative methods like LSQR (Paige & Saunders, 1982a;b) depend adversely on the problem's conditioning $\frac{\sigma_{\max}^2(\Phi)}{\lambda}$. One way to improve the convergence of methods like LSQR is to find a preconditioning matrix $M \in \mathbb{R}^{n \times n}$ where $\sigma_{\max}^2(M\Phi) \ll \sigma_{\max}^2(\Phi)$. Once a preconditioner is formed, one solves the augmented problem

$$\arg \min_{\beta} \|M\Phi\beta - My\|_2^2 + \lambda\|\beta\|_2^2$$

using just a few iterations of an iterative solver. There are a number of classical preconditioning techniques and review articles, including Wathen (2015); Benzi (2002).

LSRN (Meng et al., 2014) offers a promising method of computing a preconditioner designed for dense, overdetermined ridge regression problems. The algorithm constructs a preconditioner by subsampling columns of the matrix $\Phi$ and computing a singular value decomposition of the subsampled matrix. We believe that using a preconditioner like LSRN is a promising future research direction.

#### F.4.2 Approximate Solution via Sketching

Matrix sketching methods from the field of randomized numerical linear algebra (Drineas & Mahoney, 2016) aim to reduce the dimension of a linear system while approximately preserving the solution. This can be performed by randomly selecting rows of the (possibly preconditioned) matrix $\Phi$ and solving the smaller problem instance. Sketching algorithms specifically for overdetermined ridge regression have been introduced in recent years Chowdhury et al. (2018); Kacham & Woodruff (2022).

