# OpenReview forum: "Rotation-Invariant Random Features Provide a Strong Baseline for Machine Learning on 3D Point Clouds"
_TMLR — Accepted by TMLR_

### Review · Reviewer_cvrh · 2023-04-15

**Summary Of Contributions:**

The paper proposes a method to compute learnable rotation-invariant features for the purpose of solving rotational invariant tasks like classification of point clouds.
The core idea follows the method of random fourier features, and generates a randomly-weighted combination of volumetric functions which are a composition of radial functions and spherical harmonics functions. These functions are then used to compute features by amortizing their multiplication with the (zero centered) pointcloud function across all possible rotations. Through properties of spherical harmonics, this integral has a closed form solution. Each such operation (composed with a $\sin$ function) results in an entry inside the feature matrix $\Phi$. This feature matrix is then used to learn parameters $\beta$ through ridge regression towards matching target labels $\hat{y}$.

**Audience:**

Yes

**Claims And Evidence:**

Yes

**Requested Changes:**

* suggest practical use-cases
* comparison to more recent baselines
* Using the features in a deep net
* parameters sensitivity analysis


**Strengths And Weaknesses:**

**Strengths**:
(+) The paper tackles an important problem in 3D point cloud analysis, of designing invariant functions. The motivation is clear, the writing is well polished and concise and made self contained.

(+) The adaptation of random features to the 3D case is quite elegant and well explained


**Weaknesses**:
(-) While the paper proposes an elegant method its performance falls short compared to some reported baselines. It would be interesting to explore in which scenarios the proposed method could provide an advantage. The authors suggest that it can be used to bound the performance of more sophisticated neural network based methods however given the gaps with baselines, in which cases would that bound be useful?

(-) The proposed method involves choosing radial functions and other weight parameters, which may affect its performance. A sensitivity analysis of these parameters would provide more insights into the model's behavior and could help guide future improvements.

(-) It would be helpful to understand better why the proposed method is not as performant as some other methods. For example, is it because the invariant features are not expressive enough, or because they are used in ridge regression rather than as input to a deep net? Additionally, exploring how the method's performance would be affected if the shapes were aligned to a canonical pose, or if the amortization causes a large gap in representation capacity, could provide useful insights.

(-) The paper could benefit from a discussion of its relation to other recent rotation equivariant works. For example, Frame Averaging is a more general approach to making any function applied to the point cloud equivariant. Highlighting where the proposed method could be preferred over these alternatives would provide more context and a more nuanced understanding of its strengths and limitations.


**References**:
[1] Frame averaging for invariant and equivariant network design
[2] Vector Neurons: A General Framework for SO(3)-Equivariant Networks

---

> ### Author Response · Authors · 2023-05-07
> **Response to Reviewer cvrh 1/2**
>
> Reviewer cvrh, thank you for your review.
>
> > While the paper proposes an elegant method its performance falls short compared to some reported baselines. It would be interesting to explore in which scenarios the proposed method could provide an advantage.
>
> We agree with the characterization that our method does not offer an advantage purely in terms of test performance when compared to previously-published methods. We see the low prediction latency enjoyed by our model on sparse point clouds as an advantage over other methods. We have updated the text of the introduction to outline possible applications of our proposed method. We have copied below part of the updated introduction:
>
> “trigger algorithms in the ATLAS experiment at the CERN Large Hadron Collider require prediction latencies ranging from 2 microseconds to 40 ms at different stages of the event filtering hierarchy (Collaboration, 2008). The majority of data-driven trigger algorithms either impose rotational invariance in phase space (Komiske et al., 2018; Thaler & Van Tilburg, 2011) or enforce invariance to the Lorentz group, which includes three-dimensional rotations and relativistic boosts (Bogatskiy et al., 2022; gong et al., 2022). Another use-case for low-latency models is as replacements for force fields in molecular dynamics simulations (Gilmer et al., 2017). These calculations make serial subroutine calls to force field models, meaning latency improvements of these force field models are necessary to speed up the outer simulations.”
>
> > The authors suggest that it can be used to bound the performance of more sophisticated neural network based methods however given the gaps with baselines, in which cases would that bound be useful?
>
> We believe our method is useful because it provides a baseline which was missing from the literature. Before this work, it was unclear whether general-purpose rotation-invariant neural networks were successful due to the inductive bias of rotational invariance, or due to the inductive bias provided by deep neural networks. One way we interpret our work is showing that for certain molecular property prediction tasks, the inductive bias of rotational invariance is enough to provide very strong performance, while in 3D shape classification, deep neural networks are needed to learn hierarchical features. We have updated the text in the Introduction and Conclusion sections to make this clearer.
>
> > The proposed method involves choosing radial functions and other weight parameters, which may affect its performance. A sensitivity analysis of these parameters would provide more insights into the model's behavior and could help guide future improvements.
>
> Thank you for this suggestion. We have included a sensitivity analysis for the following: the maximum frequency of spherical harmonics, the standard deviation of the randomly-drawn weights, and the width of our radial functions. We perform this sensitivity analysis on the QM7 and ModelNet40 datasets, and results are in Appendix E. The maximum frequency of spherical harmonics and the standard deviation of the random weights are both related to the expressivity of the model. As one increases either of these parameters, the model is more expressive and is able to fit regression and classification functions to a lower training error. However, increasing the parameters too much causes the models to overfit.
>
> The sensitivity analysis showed that we were using suboptimal hyperparameters for the ModelNet40 experiment. We found a better version of the model through this analysis and updated table 3 accordingly.
>
> > It would be helpful to understand better why the proposed method is not as performant as some other methods. For example, is it because the invariant features are not expressive enough, or because they are used in ridge regression rather than as input to a deep net?
>
> Thank you for this experiment idea. We have added experiments in Appendix D which discusses the applicability of using our random features as inputs to neural networks. We evaluated which architectures were able to make predictions with reasonably low latency (< 3x the latency of random features + ridge regression), and we trained those architectures. We found that none of the low-latency architectures were able to outperform the ridge regression baseline. Simultaneously, we get zero training error in many cases, suggesting the invariant features are highly expressive. One possible reason for the performance gap is our focus on low-latency models, whereas more performant models have much longer prediction latency.

---

> > ### Author Response · Authors · 2023-05-07
> > **Response to Reviewer cvrh 2/2**
> >
> > > Additionally, exploring how the method's performance would be affected if the shapes were aligned to a canonical pose
> >
> > We discuss this question with a new paragraph in the Related Works section. We are highly doubtful about the application of a canonical alignment to molecular point clouds. To give some background, we have previously experimented with a model combining PCA-based alignment with random features in three dimensions. This method failed when applied to molecular point clouds, and we believe this is because molecular point clouds are qualitatively different from the densely-sampled point clouds from, e.g. ModelNet40. In the molecular datasets, there are orders of magnitude fewer points, so noise in the atomic locations has a larger effect on the alignment than equivalent noise applied to point clouds from shape classification tasks. Qualitatively, the alignment does not look very useful, in the sense that it does not place similar structures in similar parts of Euclidean space. We have added the following paragraph to section 5.1.1:
> >
> > “In addition to our ridge regression models, we also try using our random features as an input to deep neural networks. We were unable to find deep neural network models with reasonable prediction latencies that outperformed our ridge regression model (Appendix D). We also experimented with a model which combined a rotation-invariant PCA alignment (Xiao et al., 2020; Puny et al., 2022) with three-dimensional random features, but the alignments produced by this method were uninformative, and the model failed to meet basic baselines.”
> >
> > > or if the amortization causes a large gap in representation capacity, could provide useful insights.
> >
> > We do not believe that integrating over all possible rotations of the data creates a gap in representation capacity. We believe that our sensitivity analysis shows this – the training errors are near zero for many hyperparameter settings on the ModelNet dataset and on the QM7 dataset.
> >
> > > The paper could benefit from a discussion of its relation to other recent rotation equivariant works. For example, Frame Averaging is a more general approach to making any function applied to the point cloud equivariant. Highlighting where the proposed method could be preferred over these alternatives would provide more context and a more nuanced understanding of its strengths and limitations.
> >
> > Frame Averaging and Vector Neurons both look excellent; thank you for bringing these references to our attention. We have added the Vector Neurons result to Table 3, and we have added a paragraph describing these (and other) recent works in section 2 under the “General-purpose invariant architectures” subheading.
> >
> > While Frame Averaging and Vector Neurons are elegant ideas, we are doubtful about their applicability to molecular point clouds. Molecules are qualitatively different from the point clouds generated by computer vision applications, and we have revised the related work section to comment on these differences. The relevant text is copied below:
> >
> > “Molecular point clouds are qualitatively different [from point clouds generated by computer vision applications]; they are quite sparse, and the Euclidean nearest neighbor graph does not always match the graph defined by molecular bond structure. Because of these differences, the alignment methods and network architectures designed for solving shape classification and segmentation tasks are not a priori good models for learning functions of molecular point clouds. To the best of our knowledge, they have not been applied successfully in molecular property prediction tasks.”

---

### Review · Reviewer_RJHT · 2023-04-20

**Summary Of Contributions:**

A rotation-invariant version of the random feature method is proposed in this paper for processing point cloud data, and is applied to the tasks of molecular property prediction and 3D shape classification.

**Audience:**

Yes

**Broader Impact Concerns:**

The article has not had a broader impact statement, e.g, use for molecular property prediction.

**Claims And Evidence:**

Yes

**Requested Changes:**

Refer to Strengths And Weaknesses.

**Strengths And Weaknesses:**

The paper is well written and easy to follow. In the paper, the authors consider transformations w.r.t all possible rotations in SO(3) on the point cloud data into the random feature method proposed by Rahimi & Recht (2007) for rotation-invariant learning, and solve the problem via spherical harmonics with well theoretical derivation, making the method reliable. Experiments conducted on QM7, QM9 and ModelNet40 datasets to evaluate the proposed method in terms of accuracy and efficiency..


However, results of the proposed method on all the three datasets are less impressive. It’s suggested to include more comparisons and analyses among different methods, e.g., those listed in Spherical CNNs proposed by Cohen et al. and some recent ones for QM7 and ModelNet40 datasets, under different settings, e.g., the test settings of ‘z/z’, ‘SO(3)/SO(3)’ and ‘z/SO(3)’ in [1] for ModelNet40 dataset. I may also suggest including in Fig.3 the inference time of the results in Table 1.

Some references are missed in Sec. 5.1.2 and all the three tables. I may suggest including in the paper some more recent references.

Besides, I could find the notation explanation of  $\phi(\cdot;w_j)$  in Sec. 4.

[1] Carlos Esteves, et al., Learning SO(3) Equivariant Representations with Spherical CNNs.

---

> ### Author Response · Authors · 2023-05-07
> **Response to Reviewer RJHT**
>
> Reviewer RJHT, thank you for your review.
>
> > It’s suggested to include more comparisons and analyses … among different methods, e.g.  some recent ones for QM7
>
> We have had difficulty finding more recent references that show experiments on the QM7 dataset. We attribute this to the presence of the QM9 and GDB13 datasets; the latter is a superset of QM7. Also, deep learning has become the dominant paradigm in machine learning for computational chemistry, and we believe the focus has shifted to larger datasets because deep learning performs much better in the large-data regime. If the reviewer has any specific methods in mind, we would be grateful for a pointer.
>
> > It’s suggested to include more comparisons … ModelNet40 datasets, under different settings, e.g., the test settings of ‘z/z’, ‘SO(3)/SO(3)’ and ‘z/SO(3)’ in [1] for ModelNet40 dataset.
>
> Thank you for this suggestion; we have run these experiments and updated Table 3 accordingly. Here is a copy of the updated Table 3:
>
> | Method | z/z | SO(3)/SO(3) | z/SO(3) |
> | --- | --- | --- | --- |
> | Spherical CNNs (Esteves et al., 2018) | 0.889 | 0.869 | 0.786 |
> | SPHNet (Poulenard et al., 2019) | 0.789 | 0.786 | 0.779 |
> | Vector Neurons (Deng et al., 2021) | 0.902 | 0.859 | 0.895 |
> | Random Features (Ours) | 0.693 | 0.692 | 0.666 |
> | PointNet++ (Qi et al., 2017) | 0.918 | 0.850 | 0.284 |
>
> Thank you for mentioning our lack of reference to Esteves et al. That was an oversight, and we now include a reference in the related work section and Table 3.
>
>  > I may also suggest including in Fig. 3 the inference time of the results in Table 1.
>
> Thank you for making this suggestion. (To ensure accuracy in our reported values, we re-ran all of the QM7 regression experiments for figure 1 and table 3 using a new train/test dataset split.)
>
> The right panel of Figure 3 now contains both training times and prediction latencies.
>
>  > Some references are missed in Sec. 5.1.2 and all the three tables.
>
> Thank you for pointing this out. We have fixed this in the updated version.
>
> > I may suggest including in the paper some more recent references.
>
> We have included a new paragraph in the related works section that describes recent innovations in general-purpose rotation-invariant neural networks. This paragraph includes aligning point clouds in a rotation-invariant alignment and the broader idea of Frame Averaging (Xiao et al., 2020; Puny et al. 2022). We also discuss the idea of Vector Neurons (Deng et al., 2021). Finally, we discuss the idea that rotation-invariant functions can be written as functions of the scalar inner products between the points of a point cloud (Villar et al., 2021).
>
>  > Besides, I could find the notation explanation of 𝜙(. ; wj) in Sec. 4.
>
> Thank you for your careful reading; we have fixed this mistake in the updated version of our paper. This expression should be $\varphi(. ; g_j)$.
>
>  > The article has not had a broader impact statement, e.g, use for molecular property prediction.
>
> Thank you for pointing this out. We have added a broader impact statement, which reads: “The task of molecular property prediction is important to a wide range of applications, including the development of new pharmaceuticals, materials, and solvents. Some of these applications may be misused.”

---

### Review · Reviewer_drhY · 2023-04-24

**Summary Of Contributions:**

This paper presents a novel approach to constructing rotation-invariant features for point-cloud data using multiple basis defined by spherical harmonics functions. The proposed decomposition method is applicable to various tasks, including energy regression in physics/chemistry and point-cloud shape classification in computer vision. The authors demonstrate state-of-the-art performance in small-scale energy regression experiments and show the applicability of their approach in larger-scale energy regression experiments and shape classification. The contribution of this paper lies in introducing new mathematical tools to the field and providing promising initial results.

**Audience:**

Yes

**Broader Impact Concerns:**

Not Applicable.

**Claims And Evidence:**

Yes

**Requested Changes:**

Critical Changes:

It would be a huge benefits if the author could concern how to address the performance issue on larger scale datasets.

Another concern is although the feature is general, the performance is only attractive in one case: small-scale energy regression. I would suggest the author improve performance on modelnet40 or exploring other cases where the propose methods can be critical or useful.

Minor changes:

There are some typos in the equations or minor typos. The authors should carefully proofread the text.

**Strengths And Weaknesses:**

This maniscript studies an interesting and important problem: how to build rotation-invariant features for point-cloud data. The authors introduce new mathematical tools and show promising initial results.

The proposed algorithm performs quite well on small-scale dataset such as energy regression in QM7 dataset. This will be particularly attractive and useful for many real-world applications.

The proposed representation is quite general and task-agnostic, making it suitable for many different tasks.

On the other hand, it’s quite concering if the proposed method can be scaled up since the feature matrix depends on the number of data points. The author also mentioned the applicability of the proposed method on larger-scale dataset and result in ill-conditioned feature matrix. It remains unknown how to use it in those cases.

In Table 2, the high-performance methods are all based on neural network. I think the proposed feature can be direclty used as the input of the networks. Is there any technical difficulties in this part. It will be more convincing if the proposed feature can further improve neural network ability.

Similarly, the performance on modelnet is not very promising. Even though this particular dataset is not the focus on this paper. It is still quite far away from the current state-of-the-art method and is not convincing.

---

> ### Author Response · Authors · 2023-05-07
> **Response to Reviewer drhY**
>
> Reviewer drhY, thank you for your review.
>
> > On the other hand, it’s quite concering if the proposed method can be scaled up since the feature matrix depends on the number of data points. The author also mentioned the applicability of the proposed method on larger-scale dataset and result in ill-conditioned feature matrix. It remains unknown how to use it in those cases.
>
> We have edited the paragraph in the main part of the paper which describes the methods we have tried and insights we took from this literature, and we have added appendix F to give more details. In the appendix, we also discuss other solution methods from the numerical linear algebra literature that we leave to future work. These solution methods include preconditioning methods that are designed to accelerate the solution of dense, overdetermined, ill-conditioned ridge regression problems, which exactly fits our needs.
>
> We would also like to mention that while the number of rows and columns of our feature matrix grows with the number of data samples, the number of trainable parameters in our model is on par with the number of parameters in other models trained on the QM9 dataset. We have compiled a list of parameter counts for some of the methods listed in Table 2:
>
> | Method | Trainable Parameters |
> | --- | --- |
> | FCHL19 (Christensen et al., 2020) | 75,000 |
> | SchNet (Schutt et al., 2018) | 185,253 |
> | Random features (Ours) | 250,000 |
> | Cormorant (Anderson et al., 2019) | 299,808 |
> | PhysNet (Unke & Meuwly, 2019) | 1,293,948 |
>
> > In Table 2, the high-performance methods are all based on neural network. I think the proposed feature can be direclty used as the input of the networks. Is there any technical difficulties in this part. It will be more convincing if the proposed feature can further improve neural network ability.
>
> Thank you for this experiment idea. We have added experiments in Appendix D, which discusses the applicability of using our random features as inputs to neural networks. We evaluated which architectures were able to make predictions with reasonably low latency (< 3x the latency of random features + ridge regression), and we trained those architectures. We found that none of the low-latency architectures were able to outperform the ridge regression baseline.
>
>  > Another concern is although the feature is general, the performance is only attractive in one case: small-scale energy regression. I would suggest the author improve performance on modelnet40 or exploring other cases where the propose methods can be critical or useful.
>
> We agree with the characterization that our method does not offer an advantage purely in terms of test performance when compared to previously-published methods. We see the low prediction latency enjoyed by our model on sparse point clouds as an advantage over other methods. We have updated the text of the introduction to outline possible applications of our proposed method. We have copied below part of the updated introduction:
>
> “trigger algorithms in the ATLAS experiment at the CERN Large Hadron Collider require prediction latencies ranging from 2 microseconds to 40 ms at different stages of the event filtering hierarchy (Collaboration, 2008). The majority of data-driven trigger algorithms either impose rotational invariance in phase space (Komiske et al., 2018; Thaler & Van Tilburg, 2011) or enforce invariance to the Lorentz group, which includes three-dimensional rotations and relativistic boosts (Bogatskiy et al., 2022; gong et al., 2022). Another use-case for low-latency models is as replacements for force fields in molecular dynamics simulations (Gilmer et al., 2017). These calculations make serial subroutine calls to force field models, meaning latency improvements of these force field models are necessary to speed up the outer simulations.”
>
> We will note that after performing the sensitivity analysis suggested by reviewer cvrh, we found a better set of hyperparameters for the ModelNet40 experiments, and our test accuracy has increased by 3%.
>
>  > There are some typos in the equations or minor typos. The authors should carefully proofread the text.
>
> Thank you for your close reading of our manuscript. We have performed a close reading and found a few typos in the main body and appendix A. If any typos persist in our updated version, we would be very grateful to know.

---

### Decision · Action_Editors · 2023-06-19

**Recommendation:** Accept with minor revision

**Comment:**

This paper bears in mind the importance of rotation invariance in many vision and machine learning tasks, and study how rotation invariance plays roles in these learning tasks, especially when deep models are simultaneously used. The paper devises a general-purpose method that learns rotation invariant functions using random features. Experiments show that the method provides a rotation-invariant baseline on some tasks, and is more efficient than other alternatives.

All reviewers acknowledge that the paper addresses interesting and important problem, and the proposed solution is elegant. Reviewers are also concerned with performance of the proposed method, especially on large-scale data and tasks, how features from the proposed method can be used as input of the deep networks, and other technical and writing issues, including sensitivity analysis of radial functions and weight parameters.

Authors have revised the paper that has addressed most of these concerns. Please further revise the paper to address the remaining issues, if any, including the writing ones.


**Audience:**

People in the field of 3D vision, point cloud analysis, and other machine learning persons working on non-Euclidean data would be interested in the work.

**Claims And Evidence:**

Yes. The proposed solution is theoretically motivated and empirically verified on small-scale data and tasks.